



# On soil health and the pivotal role of proximal sensing

Yang Hu[1], Adam Cross[2,3], Zefang Shen[1], Johan Bouma[4], and Raphael A. Viscarra Rossel[1]

[1]Soil & Landscape Science, School of Molecular & Life Sciences, Faculty of Science & Engineering, Curtin University, GPO Box U1987, Perth WA 6845, Australia.
[2]School of Molecular & Life Sciences, Curtin University, GPO Box U1987, Perth WA 6845, Australia.
[3]EcoHealth Network, 1330 Beacon St, Suite 355a, Brookline, MA 02446, United States
[4]Soil Science, Wageningen University, The Netherlands.

**Correspondence:** Yang Hu (yang.hu4@postgrad.curtin.edu.au) and Raphael A. Viscarra Rossel
(r.viscarra-rossel@curtin.edu.au)

**Abstract.** Soil underpins the functioning of all terrestrial ecosystems. Sustainable soil management is crucial to preventing further degradation of the non-renewable soil resources and achieving sustainability. The soil health concept has gained popularity as a means to this end and has been integrated into the policies of many countries and supranational organisations. We need an accurate definition and scientifically robust assessment framework for effectively measuring, monitoring and managing soil

health, a framework that can effectively be communicated to the policy arena and to stakeholders. Linking soil health to the provision of ecosystem services in line with selected UN Sustainable Development Goals (SDGs) provides an effective link with the policy arena focusing on sustainable development. This is needed because lack of operational procedures to measure soil health leads to policies that ignore soils and focus on management measures. We review the literature on soil health, its conceptualisation, the current criteria for selecting indicators and thresholds, as well as the implementation of different soil

health assessment frameworks. Most published studies on soil health focus on agriculture; however, a broader perspective that includes various terrestrial ecosystems is needed. Soil health assessments should not be limited to agricultural contexts. We highlight the significant potential of advanced sensing technologies to improve current soil health evaluations, which often rely on traditional methods that are time-consuming and costly. We propose a soil health assessment framework that prioritises ecological considerations and is free from anthropogenic bias. The proposed approach leverages modern technological

advancements, including proximal sensing, remote sensing, machine learning, and sensor data fusion. This combined use of technologies enables objective, quantitative, reliable, rapid, cost-effective, scalable, and integrative soil health assessments.

## 1 Introduction

Soil is essential for ecosystem functioning and human society. Healthy soil improves water quality by enhancing infiltration, reducing erosion, and mitigating pollution (Zimnicki et al., 2020; Keesstra et al., 2021). It contributes to climate change miti-

gation by sequestering carbon, buffering soil biota from rapid environmental changes, and regulating greenhouse gas emissions ($CO_2$, $CH_4$, $N_2O$) (Lal, 2016). Soil also supports human health by providing nutrients through food, suppressing pathogens, offering medicinal resources, and aiding immune system development through exposure to environmental microbiomes (Pep-





per, 2013; Brevik et al., 2020). However, degraded or contaminated soil can harm human health through nutrient deficiencies or exposure to toxins and pathogens (Brevik et al., 2020; Oliver and Brevik, 2024).

The global importance of soil has been recognised by the United Nations' Sustainable Development Goals (SDGs), adopted in 2015. Some goals address food insecurity (SDG1/2), water scarcity (SDG6), climate change (SDG13), biodiversity (SDG15), and health (SDG3) (Bouma, 2014; Keesstra et al., 2016). SDG 15.3 explicitly aims to halt and reverse soil degradation by 2030. The concept of soil health is central to assessing soil degradation, as its indicators reflect degradation severity. Global frameworks like the UNCCD and UNFCCC also emphasise sustainable soil management and its role in carbon sequestration
(Lehmann et al., 2020).

    Despite this recognition, soil degradation remains widespread (FAO and ITPS, 2015). Soil is a non-renewable resource forming over centuries, its degradation threatens biodiversity, climate stability, human well-being, and planetary sustainability (Alexander, 1988; Doran, 1996; Lehmann et al., 2020). Agricultural expansion and deforestation exacerbate soil degradation (Dickson et al., 2021; Burrell et al., 2020), with approximately 80% of global arable land affected by desertification, erosion,
salinisation, or carbon loss (Prăvălie et al., 2021). Growing global demand for food, water, energy, and raw materials further strains soil resources (Keesstra et al., 2016). Sustainable development, as defined in the Brundtland Report, involves meeting current needs without compromising those of future generations (WCED, 1987). Sustainable soil management is urgently needed.

    Many nations have enacted soil protection policies. The EU's Soil Strategy for 2030 highlights soil contributions to ecosys-
tem services and includes initiatives like 'Living Labs and Lighthouses' to develop region-specific soil health practices (European Commission, 2021; Bouma, 2022). In the U.S., programs such as the Conservation Stewardship Program and 2018 Farm Bill incentivise practices like crop rotation, cover cropping, and rotational grazing. However, most policies focus on carbon sequestration and water quality in agricultural contexts rather than the broader, multifaceted dimensions of soil health. Australia's National Soil Strategy outlines a 20-year plan to improve soil health at a national level, extending beyond state-specific
initiatives (DAWE, 2021).

    Despite these efforts, significant challenges remain, particularly in defining, measuring, and implementing soil health assessments. Policies often prioritise management practices without addressing broader ecosystem services (Baveye, 2021; Bouma, 2021; Bouma and Scrope, 2024). Debates around soil health frequently emphasise agricultural perspectives, neglecting the ecological needs of ecosystems themselves. Societal and cultural values, while vital, complicate definitions and hinder objec-
tive, quantitative measurements (Lehmann et al., 2020; Janzen et al., 2021; Friedrichsen et al., 2021).A pragmatic focus on environmental ecosystem services can simplify assessments while maintaining relevance. Ecosystems themselves have intrinsic needs that must guide research, as highlighted in SDG 15, 'Life on Land'. Broader societal values, including cultural and aesthetic dimensions, as well as human well-being (e.g., self-determination and connectedness), are relevant but complicate definitions and measurement (Lehmann et al., 2020; Janzen et al., 2021; Friedrichsen et al., 2021). A pragmatic focus on
environmental ecosystem services simplifies assessments and enables quantitative evaluations within socioeconomic contexts while maintaining relevance (Baveye, 2021).





For soil health to serve as a practical scientific framework, it must be clearly defined and objectively measured. Effective indicators should provide insights into underlying mechanisms and support informed soil management decisions. Current methods are often outdated, expensive, and limited in scope, lacking quantitative links to outcomes (Wood and Blankinship, 2022). Advances in information technology, sensors, and artificial intelligence (AI) and machine learning offer promising solutions for rapid, precise, and cost-effective soil health assessments at appropriate scales (Viscarra Rossel et al., 2011; Shen et al., 2022; Baumann et al., 2022; Reijneveld et al., 2024). These innovations have the potential to revolutionise soil health monitoring and deepen our understanding of soil functions (Viscarra Rossel and Bouma, 2016).

Our objectives are to:

1. Analyse current views on soil health and methods for assessing it.

2. Propose procedures for developing an objective, scientifically sound, and effective framework for soil health assessment.

3. Describe the potential of soil sensing and other innovative technologies to measure indicators and enhance assessments of soil health.

## 2 Defining soil health

The evolution from 'soil fertility' to 'soil quality' and, ultimately, to 'soil health' (Bünemann et al., 2018; Lehmann et al., 2020) reflects growing scientific awareness of soil's broader functions beyond crop production. Early assessments focused on soil fertility, defined as the soil's capacity to support crop production (Patzel et al., 2000; Bünemann et al., 2018). Over time, this expanded to include soil's roles in water and air quality and contributions to plant and animal health, leading to the concept of soil quality (Mausel, 1971; Bünemann et al., 2018). Wallace first used the term 'soil health' in 1910, initially referring to soil fertility (Wallace, 1910; Brevik, 2018). By the 1990s, as understanding of soil biology and its environmental and human health roles grew, the contemporary concept of soil health emerged, encompassing soil's multifunctionality in ecosystem functions and services (Brevik, 2018; Lehmann et al., 2020; Janzen et al., 2021; Friedrichsen et al., 2021).

The terms 'soil health' and 'soil quality' are often used interchangeably but differ conceptually. Soil health refers to the current condition of a specific soil, akin to a patient's health status, while soil quality describes the expected range of health values for a given soil type, comparable to health standards for demographic groups (Bonfante et al., 2020). The analogy with human health makes "soil health" a compelling term for engaging stakeholders. Soil fertility, though narrower in scope, remains relevant in agronomic contexts as one function of soil health (Kuzyakov et al., 2020).

Figure 1 illustrates the progression and broadening scope of the soil health concept over time. Early definitions of soil health, such as "the continued capacity of a living soil to function within ecosystem boundaries to sustain biological productivity, maintain environmental quality, and promote plant, animal, and human health" (Doran and Parkin, 1994; Doran, 1996; Doran et al., 1997), remain widely applicable. Modern refinements link soil health to ecosystem services and international policy frameworks, such as the United Nations' SDGs, where soil's contributions to ecosystem services align with global sustainability goals (European Commission, 2021).



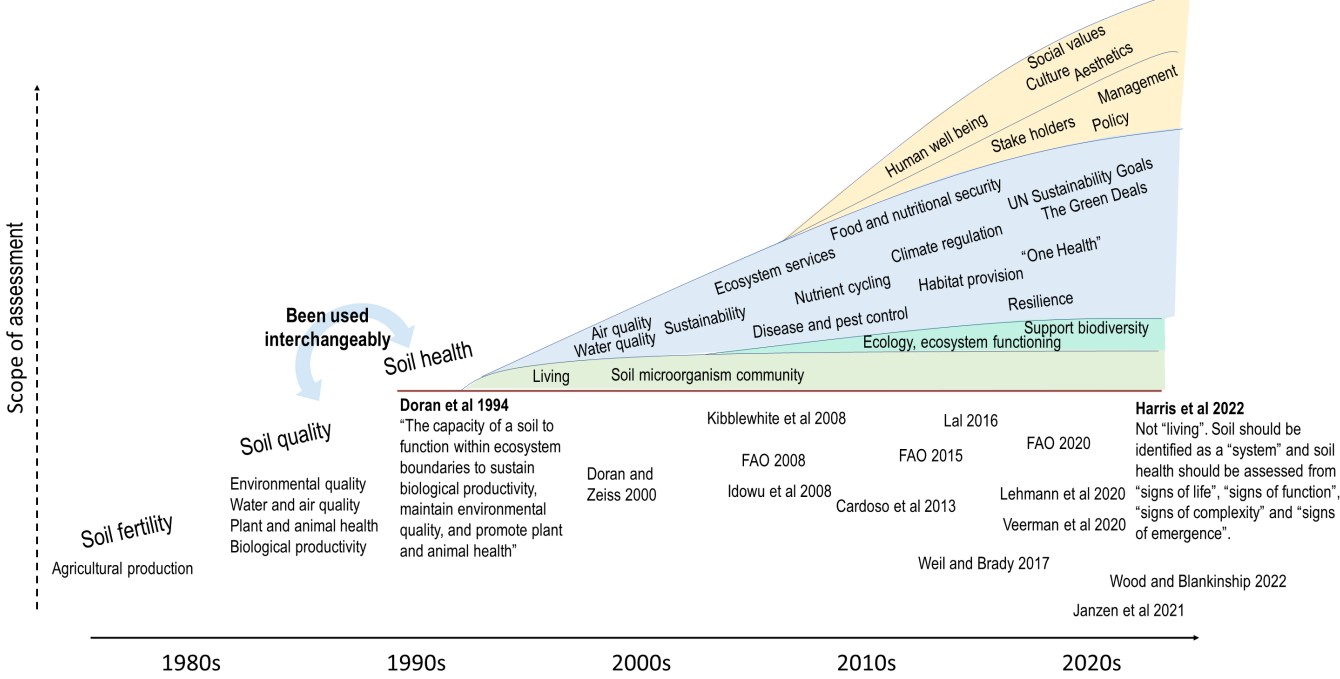

**Figure 1.** The evolution of soil assessment from soil fertility to soil quality to soil health, reflecting an expanding scope. Keywords and elements from key papers on soil health are arranged chronologically.

More recent soil health definitions emphasise soil organisms. This focus addresses the historical neglect of soil biology
compared to chemical and physical properties (Pankhurst et al., 1997) and underscores that only living organisms exhibit
health (Harris et al., 2022), a statement challenged by Bouma (2022b). Soil health is therefore described by some as 'the
biological integrity of the soil community—the balance among organisms within the soil and between soil organisms and their
physical and chemical environment' (Weil, 2017). Soil biology is a critical indicator of soil health, and there are some new
methods available for its characterisation (Reijneveld et al., 2024).

Many authors adopt a broader, holistic view of soil health, highlighting its role in ecosystem services such as food provision,
climate regulation, nutrient cycling, pest and disease control, and habitat support for soil fauna and microbiota (Lehmann et al.,
2020; Janzen et al., 2021; Friedrichsen et al., 2021). Expanding further, soil health encompasses contributions to human health,
well-being, and societal values, linking directly to the SDGs (Veerman et al., 2020). These broader definitions also integrate
stakeholder values, underscoring the need for clear, accessible descriptions and cost-effective methodologies. The soil health
concept functions as a 'boundary object', bridging knowledge and management to foster collaboration and actionable outcomes
(Wood and Blankinship, 2022). However, soil health should focus on measurable indicators rather than management practices,
as the latter aim to improve soil health and evolve with experimental findings, such as those from 'living labs' (Bouma, 2022;
Bouma and Scrope, 2024).



## 3 Limitations of Current Definition

Efforts to make soil health a holistic concept have resulted in definitions that include diverse elements such as ecosystem services, Sustainable Development Goals (SDGs), societal values, management practices, and stakeholder perspectives. This breadth, while ambitious, makes soil health difficult to measure: 'Anything that is infinitely defined is, ultimately, undefined and undefinable' (Sojka and Upchurch, 1999). Identifying and quantifying specific indicators to capture this broad scope remains a significant challenge (Baveye, 2021; Wood and Blankinship, 2022), leading to criticisms of the concept's vagueness (Janzen

et al., 2021). While some authors embrace this vagueness as an opportunity for new ideas (Janzen et al., 2021) or as a principle in itself (Lehmann et al., 2020), others argue that soil health serves better as a communication tool than a scientific framework (Powlson, 2020).

Harris et al. (2022) challenged the popular notion of soil as a 'living' entity, advocating for a holistic view that considers all soil constituents, interactions, and feedback mechanisms, rather than continually broadening the concept. Although their pro-

posal lacks a concrete implementation strategy, it underscores the importance of grounding the soil health concept in scientific understanding before setting expectations for what healthy soil can achieve. A clearer, more objective definition of soil health, aligned with scientific principles, is essential. As Lord Kelvin aptly noted, 'When you cannot measure it, when you cannot express it in numbers...you have scarcely, in your thoughts, advanced to the stage of science' (Baveye, 2021).

Current definitions often prioritise a soil's ability to provide human-desired ecosystem services over its intrinsic properties

(Kibblewhite et al., 2008; FAO and ITPS, 2015; Fine et al., 2017; Bonfante et al., 2020; Lehmann et al., 2020). However, equating soil health solely with its capacity to deliver services for humans is inadequate, much like assessing human health purely by economic output or productivity. A more effective approach is to focus on the inherent state of the soil ecosystem and its ecological functions. Ecosystem functions—representing the processes and structures within an ecosystem—provide an objective, value-neutral framework for assessing soil health, distinct from ecosystem services (De Groot et al., 2010; Manning

et al., 2018).

Some argue the soil health concept is unnecessary, suggesting that soil scientists can address issues like soil functioning, degradation, and carbon sequestration without it (Baveye, 2021). However, the concept remains valuable. It has advanced scientific understanding of soil biology (Lehman et al., 2015), highlighted its connections to human health (Brevik et al., 2020), and proved an effective tool for science communication. The term 'soil health' resonates with the public, fostering

awareness of soil science, agroecology, and conservation (Wander et al., 2019; National Resources Conservation Services of the US Dept. of Agriculture, 2019), and supports citizen science initiatives that enhance soil knowledge (Pino et al., 2022). To fully realise the potential of the soil health concept, its limitations must be addressed. Central to this is improving the ability to measure soil health. Embracing the concept requires robust methods to quantify it effectively (Bouma, 2021).

## 4 Current Soil Health Assessment Frameworks

Numerous frameworks exist for assessing soil health (Table 1). These typically involve selecting indicators, sampling guidelines, collecting and preparing soil samples, conducting laboratory analyses, interpreting results and setting thresholds, and





integrating findings into an index if desired (Rinot et al., 2019). Prominent frameworks include the Soil Management Assessment Framework (SMAF) (Andrews et al., 2004), the Comprehensive Assessment of Soil Health (CASH) (Idowu et al., 2008), and the Soil Quality Index (SQI) (Andrews and Carroll, 2001). Originally developed for soil quality evaluation, these

approaches were later adapted to assess soil health as the concept evolved (Table 1).

While the soil health concept is relevant to all terrestrial ecosystems and even aquatic and marine soils (Walden et al., 2024), most assessment frameworks focus on agriculture, aiming to identify productivity constraints and guide land managers toward sustainable practices (Andrews and Carroll, 2001; Andrews et al., 2004; FAO, 2008). Frameworks like SMAF and CASH were developed primarily using data from North American farms (Wade et al., 2022), limiting their applicability to

other regions or ecosystems. Recent efforts to expand the scope include frameworks addressing broader soil functions, such as carbon sequestration, biodiversity, and water regulation (Debeljak et al., 2019; Bonfante et al., 2020; Reijneveld et al., 2024). However, these remain largely agronomic, which is fine, if so intended, but further research is needed to broaden their applicability to other forms of land use (Wood and Blankinship, 2022).

Soil health assessments in non-agricultural systems, such as natural ecosystems, lag significantly behind, often lacking

proper thresholds that so far only apply to agriculture and forestry. Indicators could be the same but thresholds would be different.

Another limitation is scale. Most frameworks operate at the pedon or field scale, neglecting landscape-level processes such as nutrient runoff, soil redistribution, and climate change impacts (Magdoff et al., 2021; Vereecken et al., 2016). While frameworks like Su et al. (2018) address larger scales, such approaches remain rare.

Despite advancements, there is no universally accepted framework or standardised procedure for soil health assessment, particularly outside agriculture (Deel et al., 2024; Wood and Blankinship, 2022). This lack of consensus hampers the societal acceptance of soil health and its integration into ecological restoration and other disciplines (Gann et al., 2019).

## 5   Soil Health Indicators

Soil health is a multi-faceted concept that necessitates measurable indicators for assessment. These indicators—encompassing

physical, chemical, and biological properties—are closely related to ecosystem services which have an interdisciplinary character. Bünemann et al. (2018) reviewed soil health assessments and identified over 100 indicators grouped into 50 categories, with 27 frequently used across studies. This diversity complicates the selection of appropriate indicators for effective soil health evaluation. Figure 4 highlights 20 indicators: 10 physical, 12 chemical, and 8 biological.

The selection of indicators is guided by several widely accepted criteria (Bünemann et al., 2018):

1. Relevance to soil functions and ecosystem services.

2. Sensitivity to spatial and temporal variations from perturbations and management practices.

3. Practicality, affordability, and rapid measurability.

4. Reliability and reproducibility of measurements.





**Table 1.** Current soil health assessment frameworks and approaches.

| Soil Health/Quality Assessment Frameworks or Approaches | Status | Soil/Land-Use/Region Based On | Scale | Easily Adaptable to Larger Scales and Other Land-Use | Sampling and Measurement Method | Indicator Selection | Interpretation | Integration |
|---|---|---|---|---|---|---|---|---|
| Visual Soil Assessment (FAO, 2008) | Established | Agriculture | Field | No | Field test and visual assessment | Expert decision | Ordinal scale look-up table | Weighted addition |
| Soil Vital Signs (O'Neill, 2005; Amacher et al., 2007) | Established | Forest | Field | No | Sampling design reference | Expert decision | Ordinal scale look-up table | Simple addition |
| SQI (Andrews and Carroll, 2001) | Established | Agriculture | Field | No | Lab method reference | Principal component analysis | Scoring functions | Simple addition |
| SMAF (Andrews et al., 2004) | Established | Agriculture | Field | No | Not specified | Expert decision | Scoring functions | Simple addition |
| CASH (Idowu et al., 2008; Moebius-Clune, 2016; Fine et al., 2017) | Established | Agriculture | Field | No | Sampling instruction and lab method reference | Expert decision | Scoring functions | Simple averaging |
| Biofunctool (Thoumazeau et al., 2019) | Proposed & Case Study | Agriculture & Forest | Field | No | In-field and laboratory assessment | Expert decision | Scoring functions and lookup tables | Multivariate analysis weighting |
| Rinot et al (2019) | Proposed | Not specific | Not specific | Yes | Not specified | Statistical method | Scoring functions | Least square model |
| Haney Soil Health Test (Haney et al., 2018) | Established | Agriculture | Field | No | Lab method reference | Literature knowledge | Expert decision | Simple multiplication |
| Solvita Soil Health Tests (Laboratories, 2021) | Established | Agriculture | Field | No | Lab method reference | Not specified | Scored against highest expected value | Simple averaging |
| Soil Navigator Decision Support System (DSS) (Debeljak et al., 2019) | Established | Agriculture | Field | No | Not specified | Expert decision | Cognitive models | Not specified |
| Creamer et al. (2022) and BIOSIS (Zwetsloot et al., 2022) | Proposed | Not specific | Field | No | Not specified | Literature review, expert opinion, and logical sieve | Cognitive models | Not specified |
| Bonfante et al. (2020) | Case study | Agriculture | Field | Yes | Not specified | SWAT model | SWAT model | NA |
| Su et al. (2018) | Case study | Not specific | Landscape | Yes | Not specified | Literature knowledge | Mechanistic models | NA |
| Wade et al. (2022) | Proposed & Case Study | Agriculture | Landscape | Yes | Not specified | Factor analysis | Exploratory factor analysis, confirmatory factor analysis, and structural equation model | NA |
| Maaz et al. (2023) | Proposed & Case Study | Not specific | Landscape | Yes | Laboratory assessment | Principal component analysis | SEM | Latent construct from SEM |
| SEMWISE (Deel et al., 2024) | Proposed & Case Study | Not specific | National | Yes | Laboratory assessment with specified method | Not specified | SEM | Latent construct from SEM |
| Soil Health Gap (Maharjan et al., 2020) | Proposed | Agriculture | Field | No | Not specified | Not specified | Compare to 'natural', 'undisturbed' soil | Not specified |

Note: The abbreviations used are: CASH - Comprehensive Assessment of Soil Health, SEM - Structural Equation Modelling, SEMWISE - Structural Equation Model for Well-Informed Soil Evaluation, SMAF - Soil Management Assessment Framework, SQI - Soil Quality Index, SWAT - Soil & Water Assessment Tool, NA - not applicable.



5. Ability to provide actionable management insights.

Despite these criteria, no standardised method exists for selecting scientifically robust and practical indicators across differ-
ent land uses, ecosystems, and scales. Indicator selection often relies on expert judgement, which introduces subjectivity and
limits methodological transparency (Rinot et al., 2019) (Table 1). Subjectivity is further influenced by specific management
goals and the investigator's familiarity with the indicators (Wade et al., 2022). Frameworks such as SQI and CASH (Table 1)
attempt to mitigate subjectivity by using statistical techniques like principal component analysis (PCA) to identify indicators
that explain variability among land-use and management treatments (Chang et al., 2022). However, this may prioritise indica-
tors responsive to management changes while overlooking those offering unique soil health insights (Rinot et al., 2019; Wood
and Blankinship, 2022).

Methods that evaluate indicators based on specific criteria (Niemeijer and de Groot, 2008) or site-specific performance (Grif-
fiths et al., 2016; Thoumazeau et al., 2019) are valuable but often neglect interrelationships among indicators (Niemeijer and
de Groot, 2008). Improved approaches consider mechanistic interactions between indicators and the soil functions they repre-
sent (Creamer et al., 2022). For instance, the Soil Navigator decision support system (DSS) uses a hierarchical multi-criteria
model to link soil functions with sub-functions based on literature and expert insights into causal relationships among soil
properties, environmental data, land use, and management (Debeljak et al., 2019). Although primarily designed for croplands
and grasslands, this DSS informs field-scale applications (Debeljak et al., 2019). Similarly, Creamer et al. (2022) and BIOSIS
(Zwetsloot et al., 2022) (Table 1) have developed cognitive models with hierarchical structures to elucidate the relationships
between soil biota and soil processes contributing to specific functions. These models employ a logical sieve framework to
score indicators, though they currently emphasise biological indicators and require further development to integrate physical
and chemical properties (Creamer et al., 2022).

Climate change is increasingly integrated into soil health assessments, requiring indicators that address its impacts on soil
ecosystems and functions. Allen et al. (2011) identified 11 indicators for evaluating climate change effects, primarily biological,
but only four are frequently used in soil health assessments.

## 6   Measuring Soil Health Indicators

Many soil health assessment frameworks provide rudimentary soil sampling guidelines. For example, CASH (Table 1) recom-
mends combining samples from five to ten subsoil locations along a zig-zag transect (Moebius-Clune, 2016). Other frameworks
suggest general strategies, such as random sampling, W-shaped walks, or circular transects (Stott, 2019), but these methods
often fail to accurately capture soil variability. A more robust sampling strategy tailored to the assessment's specific purpose is
essential (Brus and De Gruijter, 1997). Frameworks like the one proposed by Lawrence et al. (2020) offer structured guidance
to improve soil sampling approaches.

While some frameworks recommend visual field assessments (FAO, 2008), most rely on laboratory analyses requiring sig-
nificant sample processing, such as drying, crushing, sieving, and homogenisation. Laboratory methods depend on specific
analytical equipment and standardised procedures but are often time-consuming, expensive, and procedurally complex (Vis-





carra Rossel and Bouma, 2016; Hurisso et al., 2018; Haney et al., 2018). Moreover, these methods may not accurately reflect actual soil conditions. For instance, plant-available nutrients are typically extracted using chemical reagents absent in natural soils (Haney et al., 2018), and sample preparation can disrupt soil structure, impacting assessment accuracy (Inselsbacher et al., 2011). Variations in nutrient availability due to plant strategies and environmental factors further complicate interpretations (Lambers et al., 2008). Additionally, the delay between sampling and analysis can compromise results, particularly for rapidly changing indicators like plant-available nutrients (Chen and Xu, 2008).

Laboratory testing is subject to variability both between and within laboratories (Viscarra Rossel and Bouma, 2016; van Leeuwen et al., 2022). Significant inconsistencies have been observed for routine indicators and new ones, such as permanganate oxidisable carbon (POXC), analysed by accredited laboratories or using standard methods (Hurisso et al., 2018; Wade et al., 2020). Such variability can amplify marginal errors, leading to deviations in soil health assessments and management recommendations (Viscarra Rossel and Bouma, 2016).

Logistical challenges of laboratory analyses include slow turnaround times, large costs, and environmental impact (Viscarra Rossel et al., 2011). These factors become more pronounced with increasing numbers of indicators and sample sizes (Bünemann et al., 2018; Lawrence et al., 2020). The growing demand for fine-resolution soil data across large spatial and temporal scales highlights the limitations of current laboratory methods.

## 7 Interpreting Soil Health Indicators

Conventional soil health assessment frameworks often interpret indicators using scoring curves or ordinal-scale look-up tables to generate an index value (Table 1). While ordinal-scale tables provide semi-quantitative assessments, they can introduce bias. Scoring curves transform numerical indicators into unit-less continuous values (typically 0 to 1, with 1 indicating healthy) (Wymore, 2018; Karlen and Stott, 1994). These curves are based on assumptions about the relationship between indicators and soil health outcomes, such as 'more is better', 'less is better', or 'optimum' scenarios, which may oversimplify and misrepresent complex soil dynamics (Wood and Blankinship, 2022; Maaz et al., 2023).

An alternative approach involves comparing indicator values to those from undisturbed, natural, or healthy reference sites (Maharjan et al., 2020), but defining and applying reference conditions across diverse land uses remains a challenge (Kennedy et al., 2019; Janzen et al., 2021). Conventional methods are useful for identifying differences in management practices (Stewart et al., 2018), but they often fail to establish whether these practices improve soil health or whether the indicators are sensitive enough to differentiate soil conditions (Wood and Blankinship, 2022). This underscores gaps in understanding how indicators connect to overall soil health (Creamer et al., 2022).

To address these limitations, the Soil Navigator DSS (Debeljak et al., 2019) introduced a framework that decomposes complex soil functions into sub-functions based on soil, environmental, and management interactions derived from expert knowledge and literature (Creamer et al., 2022) (Table 1). Initially developed for croplands and grasslands, this approach was later expanded to include biological indicators, with ongoing development for physical and chemical aspects (Creamer et al., 2022). More recent data-driven methods improve interpretation by analysing the covariation between indicators and latent variables



describing soil health while accounting for measurement errors (Borsboom et al., 2003; Wade et al., 2022; Deel et al., 2024). These methods simultaneously interpret indicators, focusing on structural relationships rather than predefined assumptions (Maaz et al., 2023).

Some researchers have also integrated soil health interpretation with soil-water-atmosphere-plant ecosystem models (Table 1). For example, the InVEST model assesses freshwater yield as a soil ecosystem service at landscape scales (Su et al., 2018), while the SWAP model evaluates soil health under varying climate change scenarios (Bonfante et al., 2020). These models enable systematic assessments of soil functions at broader scales through simulation (Su et al., 2018). However, understanding the complex mechanisms underlying soil health remains a significant challenge due to the intricate interactions of soil processes and functions (Vereecken et al., 2016; Vogel et al., 2023).

Threshold and target values for soil health indicators are critical for connecting indicator interpretation with management and policy (Bouma and Reijneveld, 2024). Target values represent achievable management goals, while thresholds identify critical points of soil function decline that necessitate intervention. These thresholds should be informed by scientific research (Matson et al., 2024; Agency, 2023). Reijneveld et al. (2024) emphasise separating indicators and defining threshold values to distinguish good' from not yet good enough.' Progress towards sustainability can be assessed by the number of thresholds met, with full sustainability achieved when all indicators meet their thresholds. This approach allows research to focus on indicators that fall short (Reijneveld et al., 2024).

The European Environment Agency has already defined threshold values for key indicators (Agency, 2023). A recent review summarised four methods for establishing threshold or target values: (1) using fixed values based on existing research or practical experience, (2) using values from reference sites, (3) placing indicator values within the distribution of similar soils (stratified by soil type, land use, and climate), and (4) assessing relative changes in indicator values over time (Matson et al., 2024). The relative change method, identified as the most promising, relies on representative chronosequence data, significantly increasing sampling demands (Matson et al., 2024).

Quantitative research is limited, particularly for indicators linked to multiple soil functions, such as soil organic carbon storage, and for those with complex interactions. Rapid and cost-effective methods for assessing soil health indicators are urgently needed to support the development of actionable thresholds and targets tailored to diverse soil types, ecosystems, land uses, and scales.

## 8  A Soil Health Index

Many soil health assessment frameworks (Table 1) integrate indicators into a composite soil health index to simplify communication with stakeholders. However, achieving a scientifically robust yet uncomplicated integration method remains challenging. Current approaches—such as addition (Andrews and Carroll, 2001), averaging (Moebius-Clune, 2016), multiplication (Haney et al., 2018), or weighted combinations assume linear, independent contributions of indicators to soil health, failing to account for ecosystem-specific context dependencies (Wood and Blankinship, 2022). Data-driven methods, like principal component analysis for assigning weights (Yu et al., 2018), may bias results toward indicators sensitive to management or disturbance,





overlooking those more directly linked to soil functions (Rinot et al., 2019). Emerging frameworks attempt to address these limitations using multi-criteria decision models (Debeljak et al., 2019), cognitive models (Creamer et al., 2022), and structural
equation models (Maaz et al., 2023; Deel et al., 2024) (Table 1).

A single soil health index or score is often inadequate for guiding management decisions. Individual indicators, such as soil carbon or structure, provide actionable insights, e.g., adding manure to increase carbon or adjusting tillage to improve structure, while a composite index lacks this specificity (Baveye, 2021; Powlson, 2020). Reijneveld et al. (2024) decided therefore not to define a single soil health index. Demonstrating that certain indicators don't meet their threshold allows a
focused research effort on such indicators. Possible future approaches for integrating soil health indicators into an index should balance simplicity with scientific rigor, summarising complex interactions among indicators while also offering quantitative information on individual soil properties, processes, and functions to support targeted management actions (Hussain et al., 2022).

## 9   An Ecological Focus for Soil Health

Soil health assessments often exhibit anthropogenic bias, focusing primarily on ecosystem services tied to human values, agriculture, and societal goals, including the SDGs (Figure 1, Table 1) (Kibblewhite et al., 2008; FAO and ITPS, 2015; Fine et al., 2017; Bonfante et al., 2020; Lehmann et al., 2020). This emphasis, combined with ambiguous and competing definitions, renders the concept of soil health vague and subjective (Powlson, 2020; Baveye, 2021; Janzen et al., 2021). As a result, soil health assessments remain limited and scientifically contested (Baveye, 2021). For instance the absence of standardised
procedures has led to environmental regulations that sideline soil health, instead focusing on management measures assumed to promote sustainability without clear justification (Bouma and Scrope, 2024). The scientific community cannot, therefore, afford to further delay development of operational procedures to assess and judge soil health.

We propose reorienting soil health towards a more ecological perspective, emphasising the functioning of soil systems and using robust, innovative methodologies. Figure 2 illustrates soil health within two interrelated but distinct contexts: the
ecosystem and the socio-cultural context. This perspective underscores the ecological functions of soil rather than solely its agricultural or anthropocentric roles, aiming to balance ecosystem integrity with human needs.




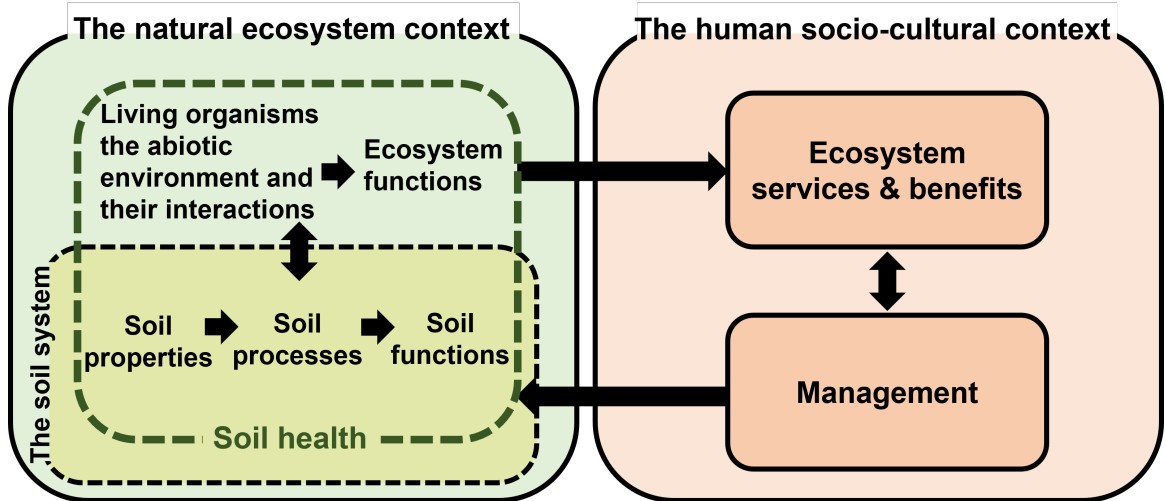

**Figure 2.** Soil health in the ecosystem and socio-cultural context.

In Figure 2, ecosystems and biodiversity–comprising living organisms, abiotic environments, and their interactions–drive ecosystem functions that underpin services and benefits for humans (De Groot et al., 2010; Manning et al., 2018). Soil, as the Earth's unconsolidated surface layer (Schoonover and Crim, 2015), is an important part ecosystem and hosts immense

biodiversity. It is a self-organising, open system that interacts dynamically with the broader biotic and abiotic ecosystem elements (Burdock and Crawford, 2015; Vogel et al., 2018).

Soil health emerges from the interplay of its physical, chemical, and biological processes (Figure 2). These processes, influenced by soil properties (e.g., organic matter, pH, texture) and environmental factors (e.g., climate, topography, biodiversity), generate essential functions such as nutrient cycling, carbon storage, water regulation, and resilience (Hoffland et al., 2020;

Gerke, 2022). For example, soil organic matter interacts with physical and biological properties to drive processes like mineralisation, aggregation, and microbial growth (Hoffland et al., 2020; Vereecken et al., 2016). Such processes integrate within ecosystems to sustain services that support both ecological and socio-cultural contexts (Figure 2).

While soil health indicators and thresholds vary across ecosystems, a value-neutral assessment framework (Figure 2) will be universally applicable. For instance, Janzen et al. (2021) demonstrated how soil health assessments in a southern Alberta

grassland produced divergent outcomes depending on perspective. Agronomic criteria deemed the soil unhealthy due to low organic matter content, alkaline pH, and thin topsoil, whereas a landscape aesthetic evaluation rated it as healthy. Both evaluations used similar indicators but applied different thresholds based on land-use context. This underscores the need to evaluate soil objectively, focusing on its intrinsic properties, processes, and functions at relevant spatial and temporal scales. By directly measuring and monitoring soil properties linked to processes and functions, we can enhance evidence-based understanding of

soil systems, inform ecosystem services and supports human interests through targeted, evidence-driven management decisions (Neßhöver et al., 2012; Elmqvist et al., 2012).





## 10 Sensing Soil Health

Quantitative and objective soil health assessments require indicators that accurately capture soil variability and adapt to diverse ecosystems and land-uses. Soil sensing technologies, used either in laboratories or in the field through proximal sensing, effectively meet these criteria. These technologies can operate independently or be combined with remote sensing to characterise broader environmental contexts, enabling the scaling of soil health assessments across larger areas (Grunwald et al., 2015).

Unlike conventional methods, sensors offer faster, more cost-efficient, and consistent ways to obtain quantitative data at higher spatial and temporal resolutions (Viscarra Rossel et al., 2011; Viscarra Rossel and Bouma, 2016; Silvero et al., 2023). Sensor-generated data often provide a more accurate representation of soil conditions compared to results from laboratory preparations and chemical extractions (Viscarra Rossel and Bouma, 2016; van Leeuwen et al., 2022; Haney et al., 2018). Moreover, certain sensors can measure multiple soil properties simultaneously or be integrated to deliver a holistic approach to soil measurement (Viscarra Rossel et al., 2011; Karlen et al., 2021).

Advances in data analysis and artificial intelligence (AI) further enhance the localisation, spatial interpretation, and temporal analysis of sensor data (Rossel et al., 2024; Teng et al., 2018; Deng et al., 2013). These capabilities position sensing technologies as promising next-generation tools for soil health assessment and monitoring (Buters et al., 2019; Viscarra Rossel and Bouma, 2016; Reijneveld et al., 2024).

## 11 Sensor-based soil health indicators

Sensing is a valuable tool for assessing and monitoring soil health, as it meets the criteria for effective indicator selection (see above). One of its key advantages is the ability to measure or estimate complementary soil properties. This capability enables cost-effective evaluation of multiple indicators, providing a more comprehensive understanding of soil processes, functions, and overall health (Figure 3). Unlike conventional methods, sensing systems are particularly adept at capturing spatial and temporal variations, enhancing their effectiveness in soil assessment.



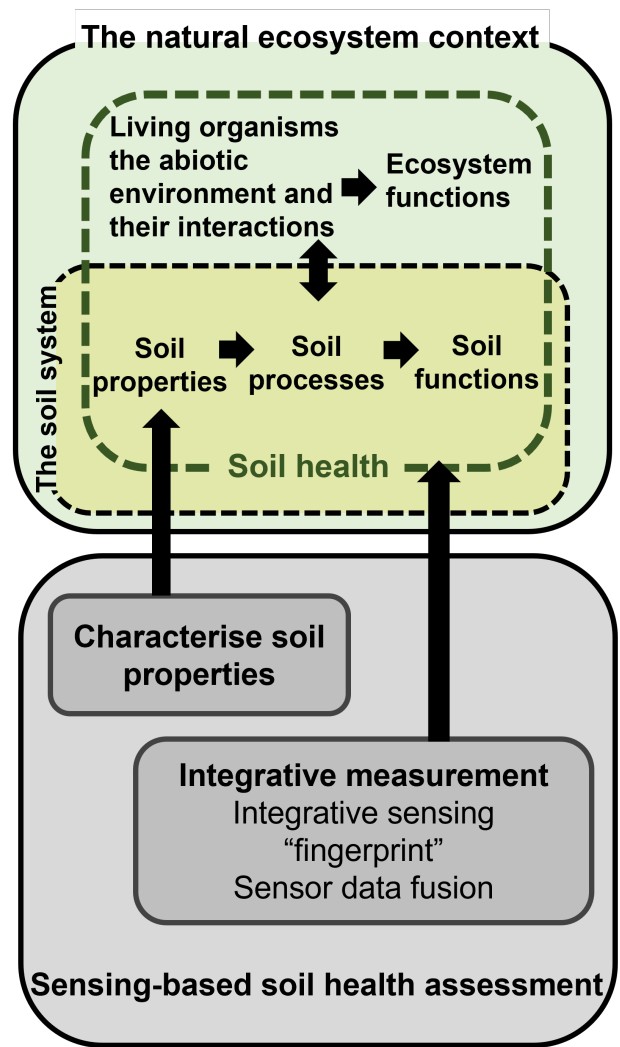

**Figure 3.** New generation sensing-based soil health assessment

While sensing may yield less precise measurements per sample than conventional methods, it compensates by allowing for a much higher volume of measurements across diverse locations and times. Such broader datasets can significantly enrich our understanding of soil variability and aid in detecting changes caused by disturbances and management practices (Elliott et al., 2007; Cécillon et al., 2009; Viscarra Rossel et al., 2010). Sensor-based measurements are typically rapid, cost-effective, and require less labour than traditional laboratory techniques. Additionally, the portability of many sensors enables in-field, proximal measurements, eliminating costs associated with sample transport, storage, and preparation (Viscarra Rossel et al., 2011). These attributes make sensing a streamlined and efficient approach for soil health assessment and monitoring (Reijneveld et al., 2024).



Sensor-based approaches have the potential to revolutionise the evaluation of soil health, especially when integrated with artificial intelligence for data interpretation and remote sensing to capture broader environmental characteristics. This combination of technologies can lead to a deeper understanding and more effective upscaling of soil health assessments, paving the way for next-generation methodologies in soil monitoring (Figure 3).

## 345  12  Sensing for characterising soil health

Various sensor technologies are available for measuring soil properties (Kuang et al., 2012; Viscarra Rossel et al., 2011; Silvero et al., 2023; Adamchuk and Rossel, 2010). These technologies include laboratory bench-top instruments and portable or in-situ proximal sensing tools. Viscarra Rossel et al. (2011) offer a thorough review and classification of soil sensors. Here, we focus on soil sensors relevant to soil health assessments, emphasising those that meet the accepted criteria for selecting indicators (see above). Figure 4 shows these sensors along with their capabilities. Although these sensors are currently used in soil science research and some specific applications (Viscarra Rossel et al., 2011; Silvero et al., 2023), standardised protocols for their use in soil health assessments are still underdeveloped.




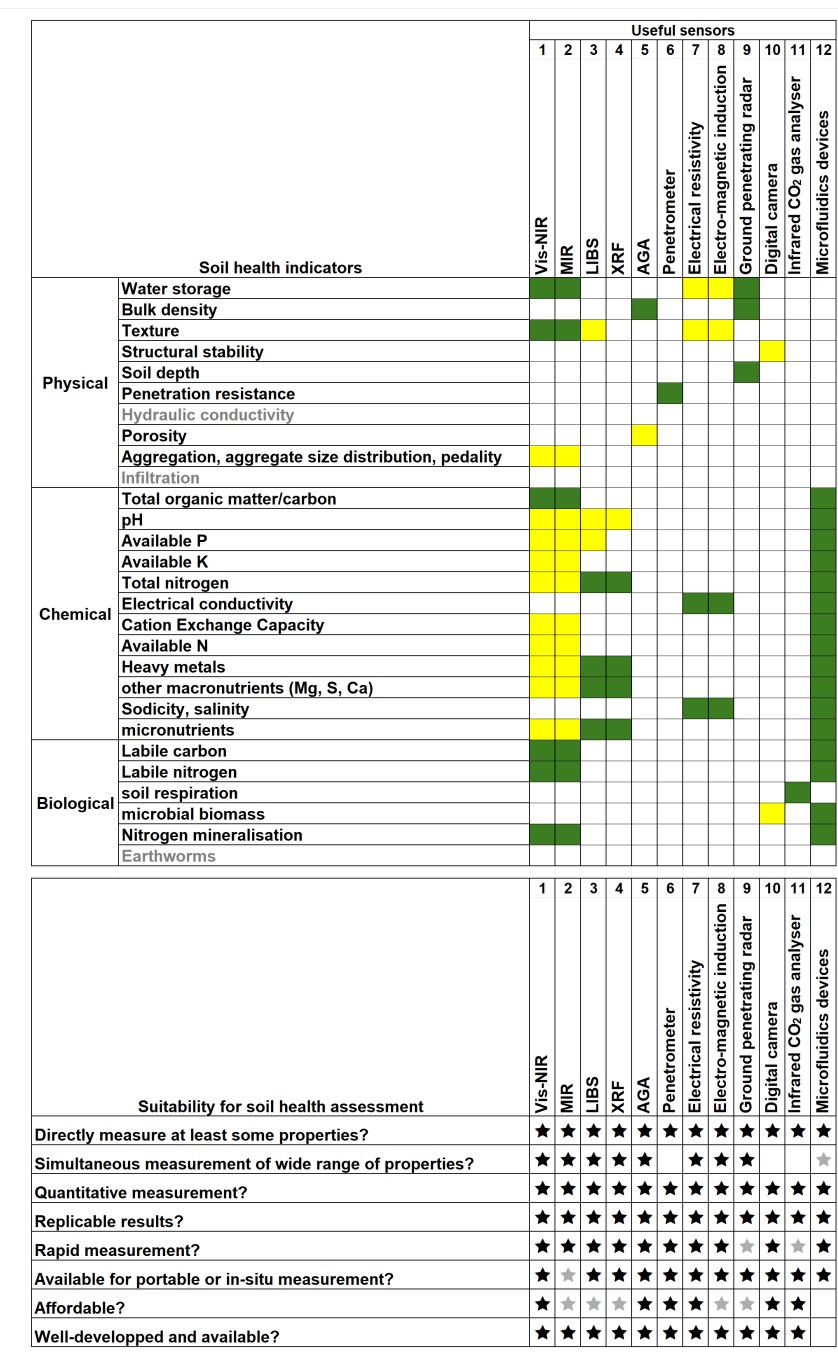

**Figure 4.** Available sensors for soil health assessment and their capabilities. The soil health indicators shown are the most frequently used, as identified by Bünemann et al. (2018). Green shades represent directly measurable soil properties, while yellow shades indicate properties indirectly measured via correlations with directly measurable ones. Grey texts indicate no suitable sensors available for these soil health indicators. (vis–NIR: Visible and near-infrared spectroscopy; MIR: Mid-infrared spectroscopy; LIBS: Laser-induced breakdown spectroscopy; XRF: X-ray fluorescence spectroscopy; AGA: Active gamma-ray attenuation).





Diffuse reflectance spectroscopy is the most mature and widely used soil sensing technology (Viscarra Rossel et al., 2022) (Figure 4), measuring organic and mineral compositions through molecular interactions with visible, near-infrared (vis–NIR),

and mid-infrared (MIR) wavelengths (Stenberg et al., 2010; Soriano-Disla et al., 2014). Spectroscopy provides results rapidly, requires little or no sample preparation, and can be applied in both laboratory and field settings. Portable, affordable versions are now available (Ji et al., 2016; Shen et al., 2022). The spectra enabling direct or indirect estimation of various chemical, physical, and biological soil health indicators (Figure 4, sensors 1 and 2) and provide a cost-effective alternative to traditional laboratory analyses (e.g. Li et al., 2022). Portable or hand-held elemental analysers, such as laser-induced breakdown spec-

troscopy (LIBS) and X-ray fluorescence spectroscopy (XRF), offer rapid, quantitative, and simultaneous measurements of soil elemental composition (Kalnicky and Singhvi, 2001; Carr et al., 2008; Senesi et al., 2009; Bricklemyer et al., 2018; John et al., 2021; Ferreira et al., 2015; Villas-Boas et al., 2016; Silva et al., 2020) (Figure 4, sensors 3 and 4).

While spectroscopy and elemental sensors measure many properties simultaneously, some critical physical, chemical, and biological properties require more specialised methods. For instance, soil bulk density, essential for assessing compaction and

estimating organic carbon stocks, can be measured using active gamma-ray attenuation (AGA) (Lobsey and Viscarra Rossel, 2016; England and Viscarra Rossel, 2018; Pepers et al., 2024) (Figure 4, sensor 5). It can also be indirectly estimated using penetrometers (Herrick and Jones, 2002), electrical conductivity sensors (Sudduth et al., 2003), or spectroscopy (Shi et al., 2023). Ground-penetrating radar measures soil depth, a critical physical indicator linked to root development, water infiltration, and nutrient availability (Sucre et al., 2011; Liu et al., 2016) (Figure 4, sensor 9). Portable respirometers with infrared

$CO_2$ analysers measure soil respiration, reflecting microbial activity and organic matter decomposition (Bekku et al., 1995; Chimner, 2004; Gyawali et al., 2020) (Figure 4, sensor 11). Soil aggregate stability, vital for preventing erosion and supporting root growth, can be assessed with digital imaging techniques such as Moulder (Flynn et al., 2020; M. Fajardo, 2023) or AS-TAVIT (Wengler et al., 2024) (Figure 4, sensor 10). Microbial biomass and fungal-to-bacterial ratios can be measured using a smartphone-based microBIOMETER (Nouri et al., 2021).

While progress has been made in sensing biological properties, further advancements are needed. Studies using vis–NIR spectra combined with machine learning have related soil spectra to microbial biomass, respiration, and bacterial and fungal abundance and diversity (Hart et al., 2020; Yang et al., 2019, 2022). Portable sequencers now facilitate efficient soil organism profiling through eDNA (Kestel et al., 2022; Hellekås, 2021), and nanobiosensors show promise for measuring enzymatic activity (Mandal et al., 2020). However, no sensors are yet available for measuring soil fauna, such as earthworms or arthropods.

Emerging technologies like microfluidics (Whitesides, 2006; Zhu et al., 2022) offer potential for analysing a wide range of soil health indicators (Figure 4, sensor 12). These "soil-on-a-chip" systems manipulate fluids in micrometre-scale channels to emulate soil environments, enabling real-time, in-situ monitoring of soil processes (Zhu et al., 2022). Combined with spectroscopy techniques, microfluidics can study interactions between soil microorganisms, the soil matrix, and plant roots, potentially advancing the development of novel, interpretable soil health indicators (Pucetaite et al., 2021). Despite their promise,

these technologies are still in early stages of development. Some soil properties remain difficult to measure with sensing, e.g., infiltration and hydraulic conductivity, and research and development is needed.



## 12.1 Sensor data fusion

No single sensor can measure all soil health indicators (Figure 3). Combining data from multiple sensors expands the coverage of soil attribute features compared to using individual sensors alone. While overlapping measurements from different sensors

may introduce some redundancy, a limited degree of overlap can enhance the robustness of sensor fusion and improve data reliability. Research has shown that integrating data from different sensors improves the predictive accuracy of models for both individual soil health indicators (Wang et al., 2015; Bricklemyer et al., 2018; Omer et al., 2020; Gozukara et al., 2022) and integrative measures of soil function and overall health (Song et al., 2024; Veum et al., 2017).

However, sensor fusion is not without challenges. Combining datasets can introduce noise, interference, and inconsistencies

(Azcarate et al., 2021), making careful planning essential. Effective sensor fusion depends on evaluating the independence of sensor information, balancing cost-effectiveness with prediction accuracy, and applying appropriate statistical methods (Azcarate et al., 2021). Furthermore, studies have found that adding data from more sensors may yield diminishing returns in model improvement, underscoring the need for judicious sensor selection (Schmidinger et al., 2024).

## 12.2 Integrative sensing

Spectroscopic sensors such as diffuse reflectance spectrometers, LIBS, and XRF (Figure 4, sensors 1 to 4) enable the simultaneous measurement of various soil constituents, including molecules, functional groups, and elements, along with their interactions. These measurements generate an integrative 'fingerprint' that can serve as a more comprehensive indicator of soil health (Cohen et al., 2005; Viscarra Rossel et al., 2006; Stenberg et al., 2010; Viscarra Rossel et al., 2022).

This integrative approach offers significant advantages for soil health assessment by mitigating the subjective and incon-

sistent selection of indicators often associated with traditional methods. Sensor signals provide a comprehensive range of quantitative soil information, minimising human bias in the indicator selection (Maynard and Johnson, 2018). This makes integrative sensing broadly applicable across diverse soil conditions, providing an innovative framework for assessing and understanding soil systems.

Integrative sensing can directly predict soil processes, functions, and health by capturing extensive soil information in a sin-

gle measurement. While promising, this capability requires further research and validation. Spectroscopic methods, especially visible-near infrared (vis-NIR) and mid-infrared (MIR) spectroscopy, have proven effective in various applications. For example, researchers have used vis–NIR spectra to predict soil C and N mineralisation (Fystro, 2002; Russell et al., 2002) and litter decomposition (Bouchard et al., 2003). Both vis–NIR and MIR spectra have been used to classify soil health, categorising soils into 'healthy', 'moderately degraded', or 'degraded', as well as to estimate soil health indices in the CASH and SMAF frame-

works (Cohen et al., 2006; Elliott et al., 2007; Maynard and Johnson, 2018; Kinoshita et al., 2012; Veum et al., 2015, 2017). Spectra have also been used to classify soil types (Viscarra Rossel and Webster, 2011; Teng et al., 2018) and predict functions such as organic carbon storage, nutrient supply, and biological activity under various conditions, including wildfire disturbances (Cécillon et al., 2009). In agriculture, spectra have been used for assessing soil fertility and evaluating carbon sequestration potential (Vågen et al., 2006; Viscarra Rossel et al., 2010; Deiss et al., 2023; Baldock et al., 2019; Karunaratne et al., 2024).



These applications highlight the versatility of spectroscopic sensing in providing integrative, scalable solutions for assessing soil health and functionality while addressing challenges in agriculture and environmental management.

## 13   Conclusions

The concept of soil health, emphasising its ecological dimensions, remains broad and challenging to define, often limited by an anthropocentric lens that overlooks its ecological dimensions. Existing soil health assessment frameworks primarily
focus on agricultural contexts, failing to adequately address the diverse land-use and ecosystem types that soils support across scales. This narrow approach constrains the potential to fully understand and manage soil's role in sustaining ecosystems. We propose adopting a broader ecological perspective that focuses on the soil's ability to function and provide ecosystem services, regardless of land use. This perspective recognises soils as dynamic components of all ecosystems, essential to maintaining ecological balance. However, traditional methods for measuring soil health indicators are often expensive, labour-intensive,
and inconsistent in representing field conditions, making them unsuitable for our comprehensive approach. We advocate for using a new generation of sensing-based soil health assessment methods to overcome these limitations. Integrating laboratory, proximal, and remote sensing technologies with AI and machine learning can provide scalable, cost-effective, and precise assessments of soil health and its contribution to ecosystem services. Our review demonstrates that this methodology is now sufficiently advanced to be implemented, offering a transformative tool for ecological soil health assessment. Despite these
advancements, the soil science community has yet to develop a universally accepted and operational framework that presents soil health as a compelling concept for policymakers, stakeholders, and the public. Without such a framework, policymakers may base environmental policies on incomplete or flawed scientific foundations. An ecological perspective, underpinned by sensing technologies, provides a unique opportunity to bridge this gap, guiding informed decision-making and fostering a deeper appreciation of the vital role of soils in sustaining ecosystems and human well-being. Time is running out, urgent
attention is needed by the scientific community.

*Author contributions.*  YH conducted the literature review and with RAVR wrote the manuscript. AC, ZS, and JB provided critical revisions to enhance the manuscript. RAVR conceptualised the review, edited the manuscript and acquired the funding for the project.

*Competing interests.*  At least one of the (co-)authors is a member of the editorial board of SOIL.

*Acknowledgements.*  RAVR thanks the Australian Government's Australia-China Science and Research Fund-Joint Research Centres (ACSRF-
JRCs) (grant ACSRIV000077) and the Australian Research Council's Discovery Projects scheme (project DP210100420) for funding. We also thank Dr Johanna Wetterlind for discussion on an earlier draft of the manuscript.





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
