# Peer review of "On soil health and the pivotal role of proximal sensing"

_EGUsphere, 2024_

## Referee Comment (RC1)

**Review Report – Manuscript: Soil Health Assessment and Proximal Sensing**

**1. Overview**

The manuscript provides a comprehensive review of soil health assessment methods using sensor technologies (proximal sensing), highlighting the significance of soil health in ecosystem functioning and policy development. The topic is both relevant and timely, particularly in the context of sustainable soil management, with important implications for agricultural practices and environmental policies. The manuscript is well-structured and covers a wide range of methodologies, policy frameworks, and technological advancements. However, several areas require clarification and further elaboration to enhance its scientific quality, clarity, and impact. Specifically, improvements could be made in the comparative analysis of different sensor technologies and the integration of practical aspects for their broader implementation. Overall, while the article is valuable, **I recommend accepting the paper with major revisions.**

**2. Strengths**

- The article provides a solid synthesis of sensor technologies used for soil health assessment, making it an essential reference for researchers and practitioners in the field.

- The authors effectively highlight the potential of these technologies to enhance agricultural productivity and sustainability, particularly in the face of increasing pressures such as climate change and soil degradation.

- *Relevance of the Topic:* Soil health is a critical issue, especially in the context of global food security and climate change. The paper emphasizes the significance of soil health monitoring in tackling these challenges. Moreover, the integration of sensors with data modeling (e.g., machine learning and AI) is a particularly forward-looking aspect of the review, as it suggests ways to refine soil health assessments beyond traditional methods.

**3. General comments**

- Some sections, particularly in the abstract and introduction, contain lengthy sentences that could be streamlined for better readability.
- Although various frameworks and sensor technologies for soil health assessment are mentioned, a more in-depth comparative analysis of the different technologies and methods would be beneficial. The authors should integrate a detailed discussion on the advantages, limitations, and specific applications of each sensor.
- The manuscript briefly touches on policies related to soil health but lacks a deeper discussion on the practical challenges of implementing these policies, particularly regarding the integration of sensor technologies in large-scale soil management. A discussion on policy gaps and recommendations for improvement would add value.
- The review highlights the potential of sensor-based methods but does not sufficiently discuss their limitations, such as cost, accessibility, and calibration challenges.
- Throughout the manuscript, terms such as "soil health," "soil quality," and "soil function" appear interchangeably. Clarifying their distinctions and ensuring consistent usage would improve coherence.

**4. Specific Comments**

*Abstract (Lines 1-16):* The abstract summarizes the study well, but it lacks an explicit mention of the originality of the approach. Adding a sentence that highlights what makes this review unique would help clarify the contribution of the article.

*Introduction (Lines 17-63):* The introduction provides a solid background on soil health but delays presenting the research problem. Consider introducing the core problem earlier.

- Lines 39-50: The discussion on soil health policies could be expanded to critically analyze their limitations.

- Lines 54-63: The transition to technological advancements is abrupt. A smoother connection explaining the shortcomings of existing assessment methods would help justify the focus on sensing technologies.

*Objectives (Lines 64-68):* The objectives are clear but could be more specific in addressing identified research gaps. Consider refining Objective 1 to specify key limitations in current assessment methods. Objective 3 should explicitly mention the practical applications of sensing technologies.

*Defining Soil Health (Lines 69-100):* The historical context is well presented but contains some redundancy. Streamlining this section would enhance readability. The manuscript presents multiple definitions of soil health but does not take a clear stance. A brief discussion on the preferred interpretation would improve coherence.

*Limitations of Current Definition (Lines 104-133):* The section provides a comprehensive overview of the challenges associated with defining soil health. However, consider streamlining the discussion to eliminate redundancy and enhance clarity. A brief critique of the varying opinions on the necessity of the soil health concept could strengthen the argument for a more objective definition.

*Current Soil Health Assessment Frameworks (Lines 134-157):* While Table 1 summarizes the different assessment frameworks, this section could be enhanced by a more critical analysis of the differences between these frameworks and their applicability to different regions or contexts.

A discussion on adapting these frameworks for non-agricultural ecosystems would also be valuable. The authors could discuss the scalability of these frameworks. Which ones are best suited for broad implementation, and which face barriers to adoption?

A clearer connection between the limitations mentioned (e.g., scale and applicability) and specific examples or case studies would enhance the depth of the discussion

*Soil Health Indicators (Lines 158-191):* This section is well-developed, but it would be helpful to add a summary table of the 20 indicators mentioned to improve readability and understanding of the key points.

*Measuring Soil Health Indicators (Lines 192-216):* The section could benefit from examples of more innovative sampling strategies beyond those mentioned, illustrating advancements in the field. A discussion on the potential consequences of inadequate sampling methods on the interpretation of soil health assessments would provide additional context to the importance of robust sampling protocols.

*Interpreting Soil Health Indicators (Lines 217-260):* The section on interpreting soil health indicators provides a solid overview but could benefit from a few adjustments:

- It would be useful to explore the extent of bias introduced by ordinal scales and how it affects interpretation across different ecosystems.
- Clarifying the limitations of assumptions, such as "more is better," could strengthen the argument, particularly in complex soil systems.
- A discussion on challenges in model calibration and data collection for large-scale applications.

*A Soil Health Index (Lines 261-278):* While discussing the limitations of composite indices, it would be beneficial to:

- Clarify how data-driven methods, like principal component analysis, can address non-linear interactions among indicators.

- Consider a multi-tiered approach that combines composite indices with individual actionable indicators for more precise management recommendations.
- Discuss how a balance between simplicity and scientific rigor can be achieved in practice, especially when communicating with stakeholders.

*An Ecological Focus for Soil Health (Lines 279-311):* It would be useful to define key terms like "soil health" and "ecosystem services" at the start of the section to avoid ambiguity. Strengthen the ecological perspective on soil health by including concrete examples of methodologies that could be applied in this framework.

*Sensing Soil Health (Lines 312-326):* The section on sensor-based technologies should delve more into the challenges such as costs, calibration, and accessibility. Discuss how these limitations can be mitigated to make such technologies more viable for widespread use. If you mention sensor fusion, briefly explain how different sensors complement one another.

*Sensor-based soil health indicators (Lines 327-344):* Include examples of soil health indicators that can be measured with sensors. This will give readers a clearer picture of the practical applications of these technologies. While the manuscript discusses the advantages of sensor-based systems, it would be helpful to explore how current limitations, such as data quality or sensor calibration, might be overcome.

*Sensing for characterising soil health (Lines 345-421):* The manuscript discusses the potential of sensor technologies but lacks details on challenges such as cost, accessibility, and data interpretation. The section mentions sensor fusion but does not clearly explain how different methods complement each other. A brief clarification would be beneficial.

*Conclusion & Future Directions (Lines 422-440):* The conclusion summarizes key points well but could end with a stronger statement on policy implications and recommendations for future research. Consider adding a discussion on the practical steps needed to operationalize the proposed framework.

**5. Suggestions for Improvement**

- **Manuscript structure :**
    - Reorganize certain sections to avoid repetition and make the article flow more smoothly. For example, a dedicated section on the challenges and limitations of sensing technologies could be added just after presenting the methods.
    - Ensure that the definitions of key terms like "soil health," "soil quality," and "soil function" are explicitly given in the introduction and consistently used throughout the article.

- **Inclusion of Real-World case studies:** The article would benefit from the inclusion of real-world case studies or examples that demonstrate how sensor technologies have been successfully implemented in different agricultural settings. Case studies would help readers understand the practical implications of using these technologies and provide insight into how challenges such as calibration, data integration, and cost can be addressed in real-world contexts.

- **Comparison Table:** As mentioned earlier, a comparative table summarizing the strengths, weaknesses, costs, and typical use cases for the different sensor technologies would be an excellent addition. This would offer readers a clearer guide to choosing the right sensor technology based on their needs.

- **Future directions:** The authors briefly mention the integration of sensors with AI, but a more extensive discussion of future trends and research opportunities would strengthen the article. For instance, exploring the potential for integrating sensor data with other environmental data

sources (e.g., satellite imagery, climate data) could provide more holistic insights into soil health.

**6. Questions for the Authors**

- **Sensor calibration and soil type variability:** How do you recommend dealing with the variability in sensor readings when applied to different soil types and environmental conditions?
- **Data integration:** What strategies do you suggest for integrating sensor data with other environmental data (e.g., weather, land use) to improve the accuracy of soil health assessments?
- **Implementation in developing regions:** Could you expand on the challenges and strategies for implementing sensor technologies in low-resource settings, especially in developing countries?

**7. Conclusion and recommendations**

The manuscript provides a valuable contribution to the discussion on the use of sensor technologies for soil health assessment. However, to enhance its scientific impact, it is essential to clarify certain technical points, provide a more in-depth comparative analysis of the different technologies, and focus on the practical aspects of implementing these technologies for soil management. I recommend a major revision before publication.

---

## Author Comment (AC1)

**Response to reviewer comments: On Soil health and the pivotal role of proximal sensing, by Hu et al.**

We thank the reviewers for their reviews and comments. Below, we provide our responses in blue text

Reviewer 2

There is lots of good information in this manuscript and it is generally well written. The biggest problem with this manuscript is that feels like it is two different papers. One idea is a review of the current lab-based assessment of soil health vs the proximal sensing of soil health. Both approaches have advantages and disadvantages, but they are both assessment framework neutral. This feels really valuable especially the discussion of how multiple sensors can be used. The other paper is about the author's suggestion for a more inclusive ecologically based soil health assessment approach. This second idea is underdeveloped and not directly related to the first idea. They provide a conceptual figure, but no actual guidance for how to do this other than "By directly measuring and monitoring soil properties linked to processes and functions." The sensor methods don't necessarily provide more direct measurements, they just make it possible to do a lot more measurements in space in time. My recommendation would be to remove the bulk of the discussion of soil health frameworks including the idea for a new framework and do a more coherent review of the measurements. For example, what are the citations that Vis-NIR or MIR can directly measure labile carbon and nitrogen or nitrogen mineralization? Finally, the paper delineates all the challenges of using the lab measurements and not any of the advantages, and then all the advantages of using sensors, but none of the challenges. A more balanced review is in order.

**Authors:** We thank the referee for taking the time to read our paper and for their conclusion that the paper provides good information and is generally well written. In our understanding, the reviewer has six primary comments and a recommendation. We address those next:

- Two papers rather than one. We respectfully disagree that this represents 'two different papers.' Our manuscript identifies the limitations of current soil health assessments and proposes a way forward. We note that the assessment is often an isolated activity in the

existing literature, with no clear links to societal applications. Linking the two is an essential element of our paper. Our proposed solution is a framework that enhances the ecological perspective, with soil sensing as the core enabling technology. Both components are essential to improve soil health assessments in the real world, and in our manuscript, they are complementary rather than separate ideas.

- We acknowledge that the ecological framing could benefit from additional development. We designed it as a conceptual framework that is adaptable and scalable to different contexts and ecosystems. We will provide text to illustrate how practitioners, politicians, and regulators might use our framework.

- Some sensors can provide direct measurements—for example, electrochemical systems for measuring soil pH and available nutrients. Some sensing systems can also measure multiple soil properties integratively, for example, pXRF, LIBS, and vis-NIR spectroscopy. We cited publications that demonstrate such sensing (p. 18, lines 411–418 in the original submitted ms). Of course, as the referee states, sensing also enables the acquisition of many more measurements, but that is not all they do. We will revise Section 12.2 to clarify these points.

- We have included the citation for vis–NIR and MIR integrative measurement of labile carbon and nitrogen mineralisation in line 411 on p18, Fystro (2002); Russell et al. (2002).

- We agree that the discussion can be improved. While laboratory methods are well-established as the current standard, we focused on highlighting the advantages of sensing, which are less well-known. We will add a discussion on the challenges of soil sensing to balance the discussion. This is indeed a key point of the paper. Introducing sensing techniques that can provide extensive data over larger areas implies a paradigm shift compared to the cumbersome laboratory techniques that have been applied until now. This way, soil health can become a truly operational item in the political environmental debate.

- As we state above, we believe that describing techniques on the one hand and discussing their applications in future environmental policy and management is the key element of our paper. Specific reviews on sensing technologies already exist (which we have cited in the manuscript, e.g., Viscarra Rossel et al. (2011); Silvero et al. (2023)), and our goal here is to discuss how these technologies can be leveraged to advance soil health assessment with a new ecological perspective: linking science with society.

L18-24. This paragraph is problematic. It starts by talking about soil health. Then it switches to the connection between soil and human health. However, it doesn't make the connection between soil health (as generally discussed) and human health. There are soil contamination issues with toxins and pathogens, but that is not typically what is measured in any soil health assessment. It's fine to discuss all the functions of soils, but don't conflate soil health assessment with soil assessment. None of the later discussion talks about pathogens or toxins, so either take this out or explain more concretely how soil health is related to human health.

**Authors:** Thank you for the comment. This paragraph aims to establish the broader importance of soil (including the effect on human health) before narrowing to soil health concepts in subsequent sections. We will revise this paragraph to better distinguish between the broad importance of soil and the specific focus on soil health assessment that follows in the manuscript. We will more clearly distinguish between the two ways in which the concept of soil health is used. It is used for communication purposes: the scientist investigates the soil as a medical doctor investigates his patients. Both use indicators. This helps explain the concept to the public. At the same time, human health can be affected by soil pollutants, and focusing on indicators and thresholds of pollutants represents another focus.

L28 Similarly, this paragraph and the next one are about soil degradation, but the authors state without any citations that soil health is central to soil degradation. Are they just the converse of each other or is soil degradation a subset of soil health or vice versa. Please explain the relationship between soil health and degradation.

**Authors:** Thank you for the comment. We will add citations to clarify the relationship between soil degradation and soil health (Food and Agriculture Organization of the United

Nations, 2025; Kraamwinkel et al., 2021). Soil degradation is associated with a decline in soil health, indicating a serious decline that can be quantified by specific indicators and corresponding thresholds. This way, the dramatic term 'degradation' becomes much clearer and transparent.

L42-43 What are "the broader, multifaceted dimensions of soil health?" Does policies in this sentence refer to the previous sentence or is this just policies in general? Is the problem that they only focus on agricultural land or only certain functions. Either add more context or take this sentence out.

**Authors:** Thank you for pointing out this sentence needs clarification. By 'broader, multifaceted dimensions of soil health' we mean all the ecological functions of soil, including biodiversity support, nutrient cycling, pest regulation, and habitat provision, across diverse ecosystems. Soil health affects all of them. The 'policies' mentioned in lines 42–43 refer to soil health policies we mentioned in the paragraph in general. Our point is that current policies primarily emphasise specific soil functions, such as carbon sequestration and water quality, within agricultural settings, rather than addressing the full scope of soil health across different land uses. We will revise this sentence to provide clearer context and eliminate the ambiguity.

L48-49 What are "ecological needs of ecosystems?"

**Authors:** We'll clarify what we meant by it in revision. We meant intrinsic ecological functions that ecosystems require (the array of biological, geochemical and physical processes that occur within an ecosystem, including nutrient cycling, habitat provision, and biodiversity support...) for self-regulating, self-sustaining, and recovering from disturbances, that are independent of human management goals.

Sections 3-4 have lots of useful information, but it isn't clear how switching to sensor based measurements addresses any of the difficulties associated with the current assessment frameworks described here.

**Authors:** Thank you again for acknowledging that our manuscript is useful. In our submitted manuscript, we outlined the difficulties of the current assessment framework and

how sensing would help to overcome those, see p14–p15 lines 312–343 (original submitted manuscript). We will ensure that this is again emphasised in the revised manuscript.

L170 One attempt to do this was published in https://doi.org/10.1016/j.soisec.2023.100084

**Authors:** Thank you for alerting us to that research. We will use it in our discussion. We acknowledge that their method is both scalable and adaptable due to the accessibility and logistical ease of the minimum dataset measurement. However, it is primarily based on the North American context and may not fully align with the specific goals or needs of other settings.

Section 6. Why is this section only about the field measurements? Aren't all these decisions about where to sample and how to standardize across equipment/users just as relevant for for sensors. MIR is a lab-based technique, so it still relies on collecting samples and processing them. For decades people have been studying variability among types of penetrometers (e.g. FRITTON, D. D.2. A STANDARD FOR INTERPRETING SOIL PENETROMETER MEASUREMENTS. Soil Science 150(2):p 542-551, August 1990). Field respiration varies from day to day and diurnally.

**Authors:** Thank you for this comment. Section 6 addresses the limitations of current soil health measurements broadly, including field sampling considerations and laboratory measurement challenges that affect conventional soil health assessments. Our discussion of sampling strategies at the beginning of this section (lines 192–197) applies broadly to soil health assessment regardless of the analytical method used. Note that much experience has been gained with sampling for fertility management, which is widely applied worldwide. We agree that sampling considerations apply to both conventional and sensing approaches (such as MIR), and we did not intend to suggest otherwise. We will clarify the scope of this section in the revision and ensure our discussion of sensing approaches adequately addresses both their capabilities and limitations regarding instrument precision where relevant. Sensors are not without measurement variability. However, they are practical and cost-effective, allowing practitioners to make an order of magnitude more measurements within the same budget compared to the more conventional analytical methods (Viscarra Rossel et al., 2022). This

increased measurement capacity leads to better characterisation of soil variability and a reduction in the estimation variance of soil properties (Viscarra Rossel et al., 2022).

Section 7 – 9 . Once again, there is useful information in this section, but how is it relevant to the review of lab vs sensor methods?

**Authors:** We thank the referee again for noting the uselessness of our paper. Section 7 identifies the challenges associated with current interpretations of soil health indicators, and Section 8 addresses the challenges of integrating these indicators. They are directly relevant to our paper because they highlight the limitations of current soil health assessments, which are essential and lead well to the proposed sensing-based soil health assessment framework. We structured the paper to first comprehensively identify the challenges across all components of soil health assessment before proposing improvements. Without discussing these current assessment limitations, the later sections on sensing approaches cannot be properly contextualised. The sensing methods we propose are specifically designed to address the challenges outlined in Sections 7 and 8. Section 9 follows and relates to our proposal that soil health assessments must have a more balanced ecological perspective.

L314-315 It is confusing whether the authors are talking about soil sensing or proximal soil sensing which some of the co-authors have been instrumental in defining (e.g. "the use of field-based sensors to obtain signals from the soil when the sensor's detector is in contact with or close to (within 2 m) the soil" Viscarra Rossel et al 2011). The title of the manuscript suggests proximal soil sensing, but then this sentence and the last paragraph of section 11 suggests that sensing can be in the laboratory too. For example, there is a much more robust history of using MIR in the lab than the field, so it would be really valuable to highlight the promising data from the field applications and for which measurements it seems to work as well as in the field as in the lab. A full discussion about the tradeoffs in data quality and cost/sample size even within the world of soil sensors would be extremely valuable and seems appropriate for a review paper.

**Authors:** Thank you for highlighting the confusion regarding our use of 'soil sensing' versus 'proximal soil sensing. We acknowledge that our manuscript discusses soil sensing in a broad

sense, encompassing both proximal (field-based) and laboratory-based approaches. To address this, we will: (i) clarify terminology throughout the manuscript to distinguish between proximal and laboratory-based soil sensing, ensuring consistency and precision, (ii) revise the title to reflect the broader scope of the paper accurately, (iii) expand the discussion to differentiate the strengths, limitations, and applications of field (proximal) versus laboratory-based sensing methods, with particular attention to MIR spectroscopy and (iv) enhance our discussion on tradeoffs and limitations as suggested.

L329 This is more of the most valuable components of sensor based methods. Many of the frameworks that the authors describe need texture, but if that can be measured in the field, that is a huge benefit.

**Authors:** Soil vis–NIR can be used to measure soil texture in situ (e.g. Zhang et al. (2017)) and we acknowledge that in-field measurement requires careful accounting of the influence of moisture. Several methods can mitigate the effects of moisture on spectra Ji et al. (2015); Wang et al. (2016).

L348-349. One of the criteria is practicality/affordability. The authors should be more explicit that using proximal sensing would be a very different approach to soil health sampling. Half of the rows of affordable in figure 4 suggest that the measurements aren't affordable. Is it just a question of the cost of technology coming down or will these always just be for research and not widely used? The traditional lab based techniques permit anyone to collect and submit a sample while the lab has the expensive equipment. There are efforts to try to make soil health sampling very inexpensive for small holder farmers to do themselves (e.g. https://doi.org/10.1016/j.geoderma.2020.114539). The proximal sensing approach would require there to be companies that had the expertise to do the field sampling got hired to do the sampling and analysis, but do is require such specialized equipment and expertise that it wouldn't be possible for the technology to be widely available.

**Authors:** We thank the reviewer for commenting on the practicality and affordability of sensing for soil health assessment. We agree that this represents a shift from traditional laboratory-based methods and will make the distinction more explicit in the revised

manuscript. The rationale for employing sensing is that it enables the collection of a much larger number of measurements across space and time at a lower cost *per sample* than conventional laboratory analysis (e.g. (Li et al., 2022)). While traditional lab techniques often involve expensive and specialised equipment, complex procedures, and significant labour costs, sensors are typically simpler to operate, increasingly portable, and capable of collecting data rapidly and sometimes directly in the field. We acknowledge the accessibility concern, particularly for smallholder farmers and resource-limited regions. However, we posit that recent and ongoing developments in sensing are making these tools more affordable, easier to use, and widely available. Some sensors also offer multi-property measurements, further increasing their cost-effectiveness. We believe that, with continued innovation and decreasing hardware costs, sensing technologies will increasingly support soil health monitoring efforts in both developed and developing contexts. As suggested by Referee 1, we propose expanding the discussion to include consideration of how these technologies could benefit smallholder farmers, particularly in low-resource settings. Thank you for the citation to support this point. Regarding Figure 4, we apologise for any confusion about the affordability indicators. Eleven of twelve sensors are marked as 'affordable' (indicated by stars), with black stars indicating higher affordability than grey stars. Only microfluidic devices are currently deemed unaffordable due to their early stage of development. We will clarify this in the figure description in our revision.

Section 12. It would be really valuable if the review could discuss all the sensors/indicators in Figure 4. That would be the most novel and valuable part of the review and most useful to soil health practitioners. It would be especially valuable if there could be a comparison of the methods that do similar indicators. For example, Vis-NIR and MIR look identical in the table, but there are advantages and disadvantages to the two approaches. It would also be valuable if there was some discussion about how "good" these measurements are. For example, there has been way more work on trying to predict SOC than nitrogen mineralization.

**Authors:** Thank you for this suggestion; however, our focus here is not on reviewing and contrasting sensing technologies. There are various other reviews already in the literature

that do this. We cited all of them (e.g., Soriano-Disla et al. (2014); Kuang et al. (2012); Viscarra Rossel et al. (2011); Silvero et al. (2023); Adamchuk and Rossel (2010)). Our objective here is to propose an ecologically centred framework for soil health assessments that has sensing at its core, presenting a link with environmental policy and society at large.

Figure 4. It is surprising to see infrared CO2 gas analyser in here. Soil respiration varies so much based on field conditions, it doesn't seem to fit with the other methods. Most of the current frameworks that do a measurement of respiration do a lab-based approach under standardized conditions. The difference between the green and the yellow is unclear. What does a direct measurement mean in this context? While the spectral techniques for SOC are based on the fact that different organic functional groups respond at particular wavelengths, the measurements is based on complicated algorithms and calibrations. This seems much less direct than combusting a sample and measuring the CO2. Similarly, the camera based techniques for structural stability really are directly measuring stability.

**Authors:** We appreciate the reviewer's comment and we will clarify these details in the revision.

- We included an infrared $CO_2$ gas analyser for soil respiration measurement, as it is the standard method for measuring soil respiration in both the lab and the field.

- 'Direct measurement' means the measurement of the soil property is made directly from the physical or chemical reaction between the sensor and the soil properties of interest. 'Indirect measurement' means the measurement is made based on the relationship between soil properties that can be directly measured by the sensor and the soil properties of interest. We will ensure that these are emphasised in the revision.

- Soil spectroscopy is a well-established soil analytical method, and so are the multivariate statistics used to extract information from the spectra. Soil spectra can cost-effectively estimate SOC. The approach is indirect, but physically based, as it relates to the soil's chemical composition. Compared to the conventional combustion method, one can measure orders of magnitude more samples at a significantly lower per-unit cost (noting also that a MIR spectrometer can cost less than one-third the

price of a C-combustion analyser, which also requires more sample handling and maintenance than the spectrometer).

- We agree that the camera-based techniques for structural stability directly measure stability. We will update this in the manuscript.

L404-405 I don't understand this sentence. Section 12 is all about the ways in which sensing is better at measuring the same indicators. It's not about new indicators. How does sensing changing the "selection of indicators."

**Authors:** Thank you for pointing out this unclear sentence. We will clarify this in the revision. Section 12 moves beyond measuring existing indicators. Specifically, what we meant by "Sensor signals provide a comprehensive range of quantitative soil information, minimising human bias in the indicator selection" is that the soil sensing signal can serve as an integrative indicator itself, incorporating multiple soil properties simultaneously without requiring pre-selection of specific properties. This reduces subjectivity in indicator selection compared to conventional methods.

L431-434 There was nothing in section 12 about integrating lab measurements and field measurements. This would be a valuable contribution to discuss which measurements are still hard to do with sensing and should be done in the lab.

**Authors:** We thank the reviewer for raising this point. We will include a brief discussion on the use of sensing in conjunction with conventional laboratory analysis as a current practical approach for assessments.

**References**

VI Adamchuk and RA Viscarra Rossel. Development of on-the-go proximal soil sensor systems. Proximal soil sensing, pages 15–28, 2010.

Food and Agriculture Organization of the United Nations. All definitions, 2025. URL https://www.fao.org/soils-portal/about/all-definitions/en/. Accessed: 2025-06-16.

Gustav Fystro. The prediction of c and n content and their potential mineralisation in heterogeneous soil samples using vis–nir spectroscopy and comparative methods. Plant and soil, 246(2):139–149, 2002.

W Ji, RA Viscarra Rossel, and Z Shi. Accounting for the effects of water and the environment on proximally sensed vis–nir soil spectra and their calibrations. European Journal of Soil Science, 66(3):555–565, 2015.

Clarisse T Kraamwinkel, Anne Beaulieu, Teresa Dias, and Ruth A Howison. Planetary limits to soil degradation. Communications Earth & Environment, 2(1):249, 2021.

B Kuang, HS Mahmood, MZ Quraishi, WB Hoogmoed, AM Mouazen, and EJ van Henten. Chapter four-sensing soil properties in the laboratory, in situ, and on-line: a review. Advances in Agronomy, 114:155–223, 2012.

Shuo Li, Raphael A Viscarra Rossel, and Richard Webster. The cost-effectiveness of reflectance spectroscopy for estimating soil organic carbon. European Journal of Soil Science, 73(1):e13202, 2022.

CA Russell, JF Angus, GD Batten, BW Dunn, and RL Williams. The potential of nir spectroscopy to predict nitrogen mineralization in rice soils. Plant and soil, 247:243–252, 2002.

Nélida Elizabet Quinõnez Silvero, José Alexandre Melo Demattê, Budiman Minasny, Nícolas Augusto Rosin, Jessica García Nascimento, Heidy Soledad Rodríguez Albarracín, Henrique Bellinaso, and Andrés Mauricio Rico Gómez. Sensing technologies for characterizing and monitoring soil functions: A review. Advances in Agronomy, 177:125, 2023.

José M Soriano-Disla, Les J Janik, Raphael A Viscarra Rossel, Lynne M Macdonald, and Michael J McLaughlin. The performance of visible, near-, and mid-infrared reflectance spectroscopy for prediction of soil physical, chemical, and biological properties. Applied spectroscopy reviews, 49(2):139–186, 2014.

Raphael A Viscarra Rossel, VI Adamchuk, KA Sudduth, NJ McKenzie, and Craig Lobsey. Proximal soil sensing: An effective approach for soil measurements in space and time. Advances in agronomy, 113:243–291, 2011.

Raphael A Viscarra Rossel, Thorsten Behrens, Eyal Ben-Dor, Sabine Chabrillat, José Alexandre Melo Demattê, Yufeng Ge, Cecile Gomez, César Guerrero, Yi Peng, Leonardo Ramirez-Lopez, et al. Diffuse reflectance spectroscopy for estimating soil properties: A technology for the 21st century. European Journal of Soil Science, 73(4):e13271, 2022.

De-Cai Wang, Gan-Lin Zhang, David G Rossiter, and Jun-Hui Zhang. The prediction of soil texture from visible–near-infrared spectra under varying moisture conditions. Soil Science Society of America Journal, 80(2):420–427, 2016.

Yakun Zhang, Asim Biswas, Wenjun Ji, and Viacheslav I Adamchuk. Depth-specific prediction of soil properties in situ using vis-nir spectroscopy. Soil Science Society of America Journal, 81(5):993–1004, 2017.

---

## Author Comment (AC2)

**Response to reviewer comments: On Soil health and the pivotal role of proximal sensing, by Hu et al.**

We thank the reviewer for their review and comments. Below, we provide our responses in blue text.

**Reviewer 1**

Some sections, particularly in the abstract and introduction, contain lengthy sentences that could be streamlined for better readability.

Although various frameworks and sensor technologies for soil health assessment are mentioned, a more in-depth comparative analysis of the different technologies and methods would be beneficial. The authors should integrate a detailed discussion on the advantages, limitations, and specific applications of each sensor.

The manuscript briefly touches on policies related to soil health but lacks a deeper discussion on the practical challenges of implementing these policies, particularly regarding the integration of sensor technologies in large-scale soil management. A discussion on policy gaps and recommendations for improvement would add value.

The review highlights the potential of sensor-based methods but does not sufficiently discuss their limitations, such as cost, accessibility, and calibration challenges.

Throughout the manuscript, terms such as "soil health," "soil quality," and "soil function" appear interchangeably. Clarifying their distinctions and ensuring consistent usage would improve coherence.

**Authors:**

- We will revise the manuscript and focus on improving readability by shortening sentences and avoiding duplication.

- Figure 4 (sensor table) compares the sensing systems that are most currently applicable for soil health assessments and their capabilities for measuring indicators. We provided comprehensive citations to other works that already provide detailed descriptions and discussions on the advantages and limitations of the technologies (e.g., Soriano-Disla

et al. (2014); Kuang et al. (2012); Viscarra Rossel et al. (2011); Silvero et al. (2023); Adamchuk and Rossel (2010)). We'll ensure that we capture the limitations of the sensors in the revision, but will try to avoid excessive repetition of what has already been reported in the literature. We also view our paper as a means to guide the author to relevant papers in the vast and overwhelming volume of literature on this particular subject.

- We provided a discussion on global policies related to soil health; however, we will improve this text to emphasise how soil sensing might affect their implementation. There is a current tendency to focus policies on management measures that are supposed to have a positive impact on soil health. However, without documentation that includes specific measurements of soil health indicators and corresponding threshold values, it is impossible to judge such measures. The principal advantage of sensing methods is the ability to generate a large amount of data in a short time at a reasonable cost. Classical laboratory methods did not allow these types of judgements, and the soil health concept remains attractive in principle but undefined. That's why the introduction of a framework focused on ecology, with sensing at its core, is essential to make soil health an operational concept with real impact in the political and societal debate.

- We will ensure that limitations and costs of the sensing systems are more clearly articulated in the revision. As indicated above, this is essential to create operational procedures that can be applied in practice, a key requirement for achieving societal and political relevance.

- It is incorrect to state that "soil health," "soil quality," and "soil function" are used interchangeably in the manuscript. In Sections 2, 9, and Figure 2, we describe each term and its relationships. Indeed, the soil science community would be well advised to avoid confusion about terminology. The use of the term 'soil health' is appealing because it allows for comparison with human health, facilitating effective communication with the general public.

Abstract (Lines 1-16): The abstract summarizes the study well, but it lacks an explicit mention of the originality of the approach. Adding a sentence that highlights what makes this review unique would help clarify the contribution of the article.

**Authors:** Thank you for the comment. We thought the abstract was clear, but we will revise it to more strongly emphasise the uniqueness and timeliness of our manuscript. In doing so, a key element will be the connection of technical aspects of the various methods, which have already been described by others, albeit less comprehensively, with the policy arena, emphasising that this modern technology is crucial for introducing an operational soil health concept to the real world.

Introduction (Lines 17-63): The introduction provides a solid background on soil health but delays presenting the research problem. Consider introducing the core problem earlier.

Lines 39-50: The discussion on soil health policies could be expanded to critically analyze their limitations.

Lines 54-63: The transition to technological advancements is abrupt. A smoother connection explaining the shortcomings of existing assessment methods would help justify the focus on sensing technologies

**Authors:** As per our previous response, we propose to revise the introduction to emphasise the uniqueness and timeliness of our manuscript. Therefore, we'd prefer to maintain our current structure in the introduction, as it is essential to establish the broader context and significance before the research aims. We will revise the text related to policy to improve it in line with what was mentioned above. When revising, we will improve the readability between all sections of the manuscript, and particularly the one mentioned.

Objectives (Lines 64-68): The objectives are clear but could be more specific in addressing identified research gaps. Consider refining Objective 1 to specify key limitations in current assessment methods. Objective 3 should explicitly mention the practical applications of sensing technologies

**Authors:** Thank you for commenting that our objectives are clear. We agree. We will be more specific in identifying current gaps and limitations while focusing on the major potential

of practical applications for the policy arena as discussed above.

Defining Soil Health (Lines 69-100): The historical context is well presented but contains some redundancy. Streamlining this section would enhance readability. The manuscript presents multiple definitions of soil health but does not take a clear stance. A brief discussion on the preferred interpretation would improve coherence.

**Authors:** Thank you for noting that the historical context is well presented. We will revise the section to improve readability. We described the progression in definitions from soil fertility to soil quality to soil health, and we believe that our stance is clear: we propose that soil health must encompass an ecological focus that emphasises ecosystem functions, not only anthropocentric services, and that modern sensing is essential to produce results in everyday practice.

Limitations of Current Definition (Lines 104-133): The section provides a comprehensive overview of the challenges associated with defining soil health. However, consider streamlining the discussion to eliminate redundancy and enhance clarity. A brief critique of the varying opinions on the necessity of the soil health concept could strengthen the argument for a more objective definition.

**Authors:** Thank you for noting that our overview is comprehensive. As mentioned previously, we will improve the writing to remove duplication. Regarding the referee's comment on the critique of varying opinions on the necessity of soil health, we have already addressed this (lines 125-132) by citing relevant literature and by clearly stating our position that the concept remains valuable despite criticisms. We provide specific examples of its contributions to science and communication. Since our paper's objective is to improve soil health assessment rather than debating the concept's fundamental necessity, an extensive discussion of this debate is outside the scope of our paper. But we certainly agree that the concept of soil health should be clearly defined as a starting point for any discussion.

Current Soil Health Assessment Frameworks (Lines 134-157): While Table 1 summarizes the different assessment frameworks, this section could be enhanced by a more critical analysis of

the differences between these frameworks and their applicability to different regions or context.

**Authors:** Table 1 provides a comprehensive comparison of eight key aspects (status, land-use context, scale, adaptability, sampling methods, indicator selection, interpretation, and integration approaches), which represents a systematic and critical analysis of the differences between the frameworks. The subsequent discussion explicitly addresses regional and contextual limitations, and we further critically examine the lack of standardisation and consensus, particularly outside agriculture. Subsequent sections on indicators, measurement, and interpretation also provide a critical analysis of these within specific frameworks. We believe that all the elements you mention are present, but agree that the text can be more specific so that we can move on to the core of the paper, namely, the ability to finally assess soil health with new techniques—a critical element for policy and society.

A discussion on adapting these frameworks for non-agricultural ecosystems would also be valuable. The authors could discuss the scalability of these frameworks. Which ones are best suited for broad implementation, and which face barriers to adoption?

**Authors:** Thank you for the comment. We believe that the current comparison and discussion on existing frameworks is sufficient. A Discussion on how existing frameworks could be adapted for non-agricultural systems will significantly lengthen and complicate the already long manuscript without enhancing our main message. Regarding scalability, Table 1 systematically addresses this through the 'Scale' and 'Easily Adaptable to Larger Scales and Other Land-Use' columns, showing that most established frameworks operate at the field scale with limited adaptability, while emerging approaches, such as those by Su et al. (2018); Wade et al. (2022) address landscape and broader scales. An additional discussion might be redundant.

A clearer connection between the limitations mentioned (e.g., scale and applicability) and specific examples or case studies would enhance the depth of the discussion

**Authors:** We tried to establish a connection between the limitations of the frameworks to specific examples, in section 4, p.6, lines 140–153 in the original submitted manuscript. We

provided specific citations to support these limitations, and some of those provide case studies. We certainly agree with the reviewer that case studies are essential for conveying intended messages.

Soil Health Indicators (Lines 158-191): This section is well-developed, but it would be helpful to add a summary table of the 20 indicators mentioned to improve readability and understanding of the key points.

**Authors:** Thank you for noting that the section on indicators is well-developed. We agree. But we disagree that an additional table would improve the paper. Figure 4 seems to serve this purpose, showing 30 indicators (10 physical, 12 chemical, and 8 biological) along with the sensors that can measure them. An additional table would be redundant.

Measuring Soil Health Indicators (Lines 192-216): The section could benefit from examples of more innovative sampling strategies beyond those mentioned, illustrating advancements in the field. A discussion on the potential consequences of inadequate sampling methods on the interpretation of soil health assessments would provide additional context to the importance of robust sampling protocols.

**Authors:** We worry that a comprehensive discussion on innovative sampling strategies would require an additional manuscript. A substantial body of literature on soil sampling for soil fertility exists, which has been widely applied worldwide, including thorough statistical analyses. This can also apply to soil health sampling to make sure that the data are representative. Our discussion already identifies the fundamental limitations of some current approaches and references robust alternatives, such as the structured guidance provided by (Lawrence et al., 2020; Brus and De Gruijter, 1997). Regarding the consequences of inadequate sampling, we noted that methods without a robust design fail to accurately capture soil variability, resulting in biased and unrepresentative data that compromises all subsequent analyses and management decisions.

Interpreting Soil Health Indicators (Lines 217-260): The section on interpreting soil health indicators provides a solid overview but could benefit from a few adjustments: It would be

useful to explore the extent of bias introduced by ordinal scales and how it affects interpretation across different ecosystems.

Clarifying the limitations of assumptions, such as 'more is better,' could strengthen the argument, particularly in complex soil systems.

A discussion on challenges in model calibration and data collection for large-scale applications.

**Authors:** Thank you for acknowledging that this section is strong. For the first point about exploring bias from ordinal scales across ecosystems, we will clarify what we mean by 'introducing bias' on line 218 and provide an appropriate citation. However, a detailed exploration of this topic across different ecosystems would require an extensive analysis that is beyond the scope of this study. Regarding the second point on clarifying limitations of scoring curve assumptions, we did touch on this and provided references to other papers that discuss this issue comprehensively (see line 222, p. 9 of the original submitted manuscript). For the third point, we think the reviewer is referring to the soil-water-atmosphere-plant ecosystem model mentioned in lines 237–242. We appreciate that the reviewer raised this point, and we will address it in the revision.

A Soil Health Index (Lines 261-278): While discussing the limitations of composite indices, it would be beneficial to: Clarify how data-driven methods, like principal component analysis, can address non-linear interactions among indicators.

Consider a multi-tiered approach that combines composite indices with individual actionable indicators for more precise management recommendations.

Discuss how a balance between simplicity and scientific rigor can be achieved in practice, especially when communicating with stakeholders.

**Authors:** Regarding the use of PCA analysis for indicator integration. We apologise for our rather clumsy phrasing. PCA is a linear method, and when used on its own, it will not address non-linearities among indicators. We will clarify this in the revision. Regarding the second point about multi-tiered approaches, we addressed this point in the discussion (p. 11, ln 271). There, we noted that individual indicators provide actionable management insights while composite indices lack specificity. This is important because it can guide research on

indicators that need improvement. For the third point about achieving balance between simplicity and scientific rigour in stakeholder communication, we thank the reviewer for raising this valuable point. We will provide further discussion on this in the revision. Simplifying is crucial for achieving acceptance of procedures in practice; however, if these procedures are not scientifically sound, efforts may backfire. Simplicity is key when communicating with stakeholders. While the scientific rigour does not have to be communicated for simplicity reasons, it should be ensured in the background as a key responsibility of the scientists.

An Ecological Focus for Soil Health (Lines 279-311): It would be useful to define key terms like "soil health" and "ecosystem services" at the start of the section to avoid ambiguity. Strengthen the ecological perspective on soil health by including concrete examples of methodologies that could be applied in this framework.

**Authors:** We defined 'soil health' early in Section 2, and elaborated in Section 3. Does the referee mean that we should repeat the definition at the start of this section, too? This does not seem necessary. The text in this section builds on the stated definition. 'Ecosystem services' are defined according to Hassan et al. (2005), which is universally accepted and adopted.

Regarding the strengthening of the ecological perspective, in the submitted manuscript, we provided an example of how our proposed framework might be applied using the southern Alberta grassland example (Janzen et al., 2021). We aimed to demonstrate how our proposed value-neutral assessment framework would address the divergent outcomes resulting from different evaluative perspectives. Perhaps this was unclear? In the revision, we will ensure that these parts are enhanced to clearly articulate the application of our framework.

Sensing Soil Health (Lines 312-326): The section on sensor-based technologies should delve more into the challenges such as costs, calibration, and accessibility. Discuss how these limitations can be mitigated to make such technologies more viable for widespread use. If you mention sensor fusion, briefly explain how different sensors complement one another.

**Authors:** We've responded to this comment already. Please see above.

Sensor-based soil health indicators (Lines 327-344): Include examples of soil health indicators that can be measured with sensors. This will provide readers with a clearer understanding of the practical applications of these technologies. While the manuscript discusses the advantages of sensor-based systems, it would be beneficial to explore how current limitations, such as data quality or sensor calibration issues, can be addressed.

**Authors:** We provided soil health indicators that can be measured with sensors in Figure 4. Again, we've responded to this comment before. We will revise to emphasise the challenges of the sensing-based approach.

Sensing for characterising soil health (Lines 345-421): The manuscript discusses the potential of sensor technologies but lacks details on challenges such as cost, accessibility, and data interpretation. The section mentions sensor fusion but does not clearly explain how different methods complement each other. A brief clarification would be beneficial

**Authors:** The indeed important aspects of cost, accessibility and data interpretation have been discussed above, and we have reacted to it.

Conclusion & Future Directions (Lines 422-440): The conclusion summarizes key points well but could end with a stronger statement on policy implications and recommendations for future research. Consider adding a discussion on the practical steps needed to operationalize the proposed framework.

**Authors:** Thank you for acknowledging that our conclusion concludes well. As suggested, we will revise it to strengthen our final messages. What is needed now is real action in the real world. The new sensing techniques represent a paradigm shift, finally allowing the operationalisation of the soil health concept to the benefit of the effectiveness of environmental policy and, therefore, for society at large.

Suggestions for Improvement: Manuscript structure : o Reorganize certain sections to avoid repetition and make the article flow more smoothly. For example, a dedicated section on the challenges and limitations of sensing technologies could be added just after presenting the methods. o Ensure that the definitions of key terms like 'soil health,' 'soil quality,' and 'soil

function' are explicitly given in the introduction and consistently used throughout the article

**Authors:** Thank you for summarising your review. We have thoroughly addressed the comments made.

Inclusion of Real-World case studies: The article would benefit from the inclusion of real-world case studies or examples that demonstrate how sensor technologies have been successfully implemented in different agricultural settings. Case studies would help readers understand the practical implications of using these technologies and provide insight into how challenges such as calibration, data integration, and cost can be addressed in real-world contexts.

**Authors:** We appreciate the suggestion. We will provide an example in the revision to offer readers guidance on how to implement the framework.

Comparison Table: As mentioned earlier, a comparative table summarizing the strengths, weaknesses, costs, and typical use cases for the different sensor technologies would be an excellent addition. This would offer readers a clearer guide to choosing the right sensor technology based on their needs.

**Authors:** The information is in the submitted Figure 4, but as we have written above, we will revise according to the comments made.

Future directions: The authors briefly mention the integration of sensors with AI, but a more extensive discussion of future trends and research opportunities would strengthen the article. For instance, exploring the potential for integrating sensor data with other environmental data sources (e.g., satellite imagery, climate data) could provide more holistic insights into soil health

**Authors:** We agree that our discussion on this was a little brief, and we will revise to enhance these aspects. However, thorough reviews on sensors and AI, as well as the integration of sensing with remote sensing, already exist, and we have cited them, for example, (Silvero et al., 2023; Grunwald et al., 2015; Rossel et al., 2024).

Questions for the Authors:

Sensor calibration and soil type variability: How do you recommend dealing with the

variability in sensor readings when applied to different soil types and environmental conditions?

Data integration: What strategies do you suggest for integrating sensor data with other environmental data (e.g., weather, land use) to improve the accuracy of soil health assessments?

**Authors:** Thank you for the questions. We politely suggest that these aspects are somewhat peripheral to our manuscript and have been addressed in other literature on specific sensor systems and technical reviews on sensing. Some examples of integrating environmental data with sensing are presented in (Yang et al., 2019, 2022) and (Viscarra Rossel et al., 2010). There are others, of course. We suggest the reviewer seek out the suggested (and other) literature, or contact us separately, and we'd be glad to discuss and help.

Implementation in developing regions: Could you expand on the challenges and strategies for implementing sensor technologies in low-resource settings, especially in developing countries?

**Authors:** This is a good point. We agree that implementing sensing technologies in low-resource settings is highly relevant and will add a thorough discussion on how we might use our framework in such settings. Thank you for the comment.

7. Conclusion and recommendations

The manuscript provides a valuable contribution to the discussion on the use of sensor technologies for soil health assessment. However, to enhance its scientific impact, it is essential to clarify certain technical points, provide a more in-depth comparative analysis of the different technologies, and focus on the practical aspects of implementing these technologies for soil management. I recommend a major revision before publication.

**Authors:** Thank you for acknowledging the value of our manuscript. We will certainly revise according to the comments and our responses above. Thank you.

**References**

VI Adamchuk and RA Viscarra Rossel. Development of on-the-go proximal soil sensor systems. Proximal soil sensing, pages 15–28, 2010.

DJ Brus and JJ De Gruijter. Random sampling or geostatistical modelling? choosing between design-based and model-based sampling strategies for soil (with discussion). Geoderma, 80 (1-2):1–44, 1997.

Sabine Grunwald, Gustavo M Vasques, and Rosanna G Rivero. Fusion of soil and remote sensing data to model soil properties. Advances in Agronomy, 131:1–109, 2015.

Rashid Hassan, Robert Scholes, and Neville Ash, editors. Ecosystems and Human Well-being: Current State and Trends, volume 1 of The Millennium Ecosystem Assessment Series. Island Press, 2005. ISBN 1-55963-227-5 1-55963-228-3.

H Henry Janzen, David W Janzen, and Edward G Gregorich. The 'soil health'metaphor: Illuminating or illusory? Soil Biology and Biochemistry, 159:108167, 2021.

B Kuang, HS Mahmood, MZ Quraishi, WB Hoogmoed, AM Mouazen, and EJ van Henten. Chapter four-sensing soil properties in the laboratory, in situ, and on-line: a review. Advances in Agronomy, 114:155–223, 2012.

Patrick G Lawrence, Wayne Roper, Thomas F Morris, and Karl Guillard. Guiding soil sampling strategies using classical and spatial statistics: A review. Agronomy Journal, 112 (1):493–510, 2020.

Raphael A Viscarra Rossel, Zefang Shen, Leonardo Ramirez Lopez, Thorsten Behrens, Zhou Shi, Johanna Wetterlind, Kenneth A Sudduth, Bo Stenberg, Cesar Guerrero, Asa Gholizadeh, et al. An imperative for soil spectroscopic modelling is to think global but fit local with transfer learning. Earth-Science Reviews, page 104797, 2024.

Nélida Elizabet Quinõnez Silvero, José Alexandre Melo Demattê, Budiman Minasny, Nícolas Augusto Rosin, Jessica García Nascimento, Heidy Soledad Rodríguez Albarracín, Henrique Bellinaso, and Andrés Mauricio Rico Gómez. Sensing technologies for characterizing and monitoring soil functions: A review. Advances in Agronomy, 177:125, 2023.

José M Soriano-Disla, Les J Janik, Raphael A Viscarra Rossel, Lynne M Macdonald, and Michael J McLaughlin. The performance of visible, near-, and mid-infrared reflectance spectroscopy for prediction of soil physical, chemical, and biological properties. Applied spectroscopy reviews, 49(2):139–186, 2014.

Changhong Su, Huifang Liu, and Shuai Wang. A process-based framework for soil ecosystem services study and management. Science of the total Environment, 627:282–289, 2018.

Raphael A Viscarra Rossel, Rodnei Rizzo, Jose Alexandre Melo Demattê, and T Behrens. Spatial modeling of a soil fertility index using visible–near-infrared spectra and terrain attributes. Soil science society of America journal, 74(4):1293–1300, 2010.

Raphael A Viscarra Rossel, VI Adamchuk, KA Sudduth, NJ McKenzie, and Craig Lobsey. Proximal soil sensing: An effective approach for soil measurements in space and time. Advances in agronomy, 113:243–291, 2011.

Jordon Wade, Steve W Culman, Caley K Gasch, Cristina Lazcano, Gabriel Maltais-Landry, Andrew J Margenot, Tvisha K Martin, Teal S Potter, Wayne R Roper, Matthew D Ruark, et al. Rigorous, empirical, and quantitative: a proposed pipeline for soil health assessments. Soil Biology and Biochemistry, page 108710, 2022.

Yuanyuan Yang, Raphael A Viscarra Rossel, Shuo Li, Andrew Bissett, Juhwan Lee, Zhou Shi, Thorsten Behrens, and Leon Court. Soil bacterial abundance and diversity better explained and predicted with spectro-transfer functions. Soil Biology and Biochemistry, 129:29–38, 2019.

Yuanyuan Yang, Zefang Shen, Andrew Bissett, and Raphael A Viscarra Rossel. Estimating soil fungal abundance and diversity at a macroecological scale with deep learning spectrotransfer functions. Soil, 8(1):223–235, 2022.